# Achieving large thermal hysteresis in an anthracene-based manganese(II) complex via photo-induced electron transfer

Ji-Xiang Hu [1], Qi Li[1], Hai-Lang Zhu[2], Zhen-Ni Gao[1], Qian Zhang[1], Tao Liu [2✉] & Guo-Ming Wang [1✉]

Achieving magnetic bistability with large thermal hysteresis is still a formidable challenge in material science. Here we synthesize a series of isostructural chain complexes using 9,10-anthracene dicarboxylic acid as a photoactive component. The electron transfer photochromic $Mn^{2+}$ and $Zn^{2+}$ compounds with photogenerated diradicals are confirmed by structures, optical spectra, magnetic analyses, and density functional theory calculations. For the $Mn^{2+}$ analog, light irradiation changes the spin topology from a single $Mn^{2+}$ ion to a radical-$Mn^{2+}$ single chain, further inducing magnetic bistability with a remarkably wide thermal hysteresis of 177 K. Structural analysis of light irradiated crystals at 300 and 50 K reveals that the rotation of the anthracene rings changes the Mn1–O2–C8 angle and coordination geometries of the $Mn^{2+}$ center, resulting in magnetic bistability with this wide thermal hysteresis. This work provides a strategy for constructing molecular magnets with large thermal hysteresis via electron transfer photochromism.

[1] College of Chemistry and Chemical Engineering, Qingdao University, Shandong 266071, P. R. China. [2] State Key Laboratory of Fine Chemicals, Dalian University of Technology, Dalian 116024, P. R. China. ✉email: liutao@dlut.edu.cn; gmwang_pub@163.com

Magnetically bistable multifunctional materials attracted significant attention owing to their potential applications in switches, sensors, and information storage[1–5]. Magnetically bistable phenomena with thermal hysteresis have been observed in spin-crossover materials[6–8], metal-to-metal charge transfer compounds[9–11], valence tautomeric complexes[12,13], reversible dimerization of organic π-radicals[14–16], and complexes with dynamic coordination environments[17–19]. Although bistable systems have been extensively studied, there are still some challenges. There is a need to explore wide thermal hysteresis and develop new mechanisms for constructing magnetically bistable complexes for both fundamental research and practical applications.

The width of thermal hysteresis is vital in bistable materials. Until now, only five complexes have shown large and reproducible thermal hysteresis loops with repeated scanning ($\Delta T > 100$ K)[20–24]. To achieve the broad thermal hysteresis, strengthening the cooperative interactions between magnetic molecules and extending dimensionality by π–π stacking and hydrogen bonding interactions are becoming a promising strategy[25–32]. Besides the widespread spin-crossover and metal-to-metal charge transfer compounds, valence tautomeric complexes with radicals or reversible dimerization of organic π-radicals provide an effective way for magnetic bistability with relatively wide thermal hysteresis loops ($\Delta T > 100$ K). Thus, introducing radicals into molecular magnets may be a good technique for designing systems with wide thermal hysteresis loops.

To date, direct syntheses of stable radicals are time-consuming and include unsafe redox reactions triggered by chemical and electrochemical reactions[33–35]. In contrast, owing to the rapid response and convenience of switches, electron transfer photochromic materials with light-actuated stable radicals have attracted considerable attention[36–40]. After ultraviolet–visible (UV–Vis) light irradiation (ca. 300–800 nm), stable radical analogs are generated via electron transfer and can provoke magnetic interactions with paramagnetic metal centers, further intriguing amazing magnetic behavior, such as photodemagnetization, magnetic transitions, and single-molecule magnetism[41–44]. Furthermore, such room-temperature photochromism provides an opportunity to realize the on/off switch of photomagnetic properties with color changes. According to recent reports, anthracene-based ligands exhibit radical-actuated photochromic behavior, in which ligand with π-conjugations can kinetically stabilize the photogenerated radicals[45,46], and magnetic interactions operate between radicals and paramagnetic metal ions. However, magnetic thermal hysteresis is still challenging in electron transfer photochromic materials.

In this work, we report that 9,10-anthracene dicarboxylic acid (H$_2$ADC) can act as a photoactive ligand. The reaction of H$_2$ADC with transition metal ions yields a series of isostructural single-chain complexes of $[M(ADC)(H_2O)_2(DMF)_2]_n$ (ADC = 9,10-anthracenedicarboxylate, DMF = N, N-Dimethylformamide; M = $Mn^{2+}$, $Zn^{2+}$, $Ni^{2+}$ and $Co^{2+}$ for **1**, **2**, **3**, and **4**, respectively). Photochromism is not observed in **3** and **4**, whereas **1** and **2** show visible photochromic phenomena at ambient conditions due to the generation of stable diradicals after Xenon (Xe)-lamp illumination. Originated from the variations of magnetic coupling between $Mn^{2+}$ ions and photogenerated radicals, compound **1** exhibits a remarkable photomagnetic response after light irradiation, resulting in an unexpectedly wide thermal hysteresis with a temperature width of 177 K ($T_{1/2\downarrow} = 62.7$ K; $T_{1/2\uparrow} = 239.8$ K; $T_{1/2\downarrow}$ and $T_{1/2\uparrow}$ represent the transition temperatures in the cooling and heating processes, respectively, during direct-current magnetic susceptibility measurements). We realize a magnetic thermal hysteresis loop in radical-actuated photochromic materials, and the hysteresis loop of 177 K is significantly wide compared to those reported for reproducibly bistable magnetic systems[20–24].

## Results

**Characterization of crystal structures**. Single-crystal X-ray analyses revealed isomorphism in **1–4** with the $P2_1/c$ monoclinic space group (Supplementary Tables 1 and 2). For clarity, only the Mn analog is discussed herein. In **1**, a crystallographically independent $Mn^{2+}$ center is coordinated by two DMF molecules, two ADC anions and two water molecules (Fig. 1a), forming a nearly ideal [MnO$_6$] octahedron with 0.047 (continuous shape measure analyses (CshM) by Shape 2.0, Supplementary Table 10). The Mn–O bond distance ranges from 2.1566(14) to 2.2162(18) Å, which is characteristic of $Mn^{2+}$ ions. The ADC bridges two $Mn^{2+}$ ions to construct a one-dimensional chain (Fig. 1b), and the interchain hydrogen bonding interactions between the coordinated water molecules and ADC units appear with the O⋯O distances in the range of 2.677(3)–2.754(3) Å (Fig. 1c and Supplementary Table 11), which further forms a two-dimensional (2D) architecture (Supplementary Fig. 1) by the hydrogen bonding interactions. Notably, the nearest intrachain and interchain Mn⋯Mn distances are 11.5687(9) and 7.4467(5) Å, respectively, indicating negligibly small magnetic interactions between the $Mn^{2+}$ ions. Supplementary Fig. 2 shows the structures of **2–4**, and Supplementary Tables 7–9 list their bonds and angles.

**Photochromic properties**. Powder X-ray diffraction (PXRD) was measured for all the compounds, and the experimental curves are in good agreement with the simulated one (Supplementary Figs. 3–7), indicating high purity crystal phases in these series of compounds. The electron transfer photochromic behavior in the compounds was explored under ambient conditions. When **1** is exposed to a 250-W Xe-lamp (320–780 nm), color changes from colorless to pink and white to yellow are observed in the crystalline and powder samples, respectively (Fig. 2a, insert and Supplementary Fig. 8; the colored samples are labeled as **1a**). The time-dependent solid UV–Vis spectra for the samples show a broad absorption around 475 nm upon irradiation by the Xe-lamp (Fig. 2a), which is characteristic of ADC• radicals during electron transfer[47]. The rate of photochromism is estimated from the diffuse-reflectance spectra with a half-time ($t_{1/2}$) of ~9.20 min (Supplementary Fig. 9). Compound **2** shows similar photochromic behavior as **1**, in which color changes are more pronounced (Fig. 2b, insert and Supplementary Fig. 10; photo-irradiated **2** is denoted as **2a**). Compared with **1**, the photochromic rate for **2** with $t_{1/2}$ of 12.31 min (Supplementary Fig. 11) indicates a relatively slower coloration process. Notably, the temperature dependence of the crystal structures for **1**, and the PXRD and infrared (IR) spectra for **1** and **2** before and after the light irradiation remained unchanged, indicating no isomerization or/and photodimerization (Supplementary Tables 3–6, Supplementary Figs. 3–5, and 12). In situ photoluminescence spectra for **1** and **2** were recorded with the excitation of 283-nm light, and the fluorescence intensities at 422, 445, and 473 nm, ascribed to H$_2$ADC ligands[48], decrease upon illumination by a Xe-lamp (Fig. 2c, d), suggesting that the formed ADC• radicals quench the photoluminescence. Notably, **3** and **4** show no photochromism, even after 10-h irradiation by the Xe-lamp (Supplementary Figs. 13–16). This is attributed to the overlapped absorption bands of $d$-$d$ transitions and photogenerated radicals (Supplementary Fig. 15).

Solid-state X band electron spin resonance (ESR) measurements for **1** and **2** were conducted at room temperature to further understand the photochromism. Compound **1** shows a broad resonance absorption band at $g = 2.059$ (Fig. 2e), attributed to $Mn^{2+}$ ions in the octahedral coordination geometry[49]. After continuous light irradiation for 3 h, the peak intensity decreases,

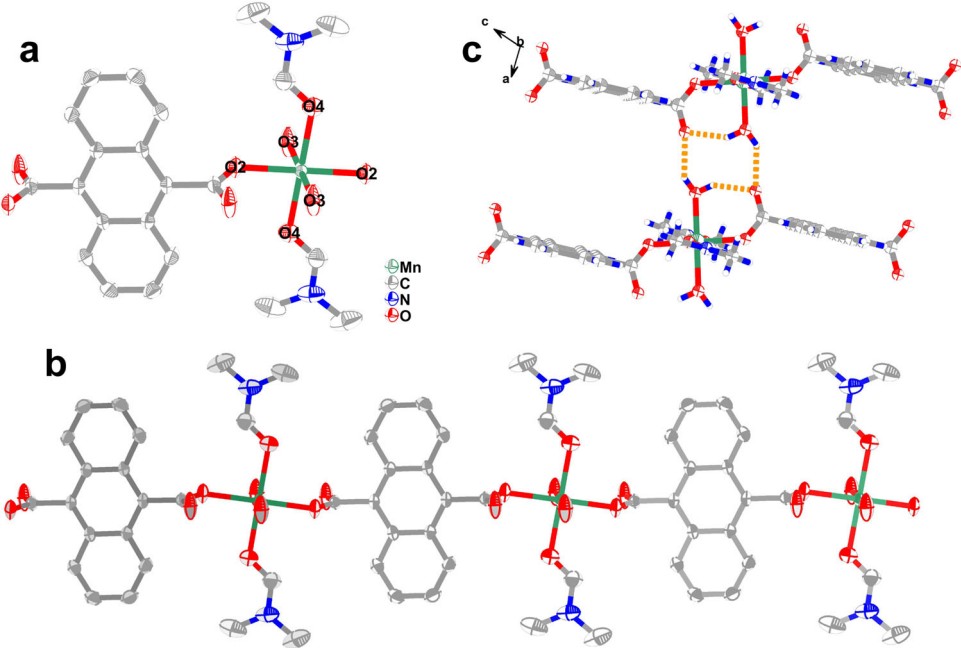

**Fig. 1 ORTEP drawings of crystal structure 1 with thermal ellipsoids set at 50% probability. a** molecular structure for compound **1**; **b** the chain structure for **1**; **c** H-bonding interactions for **1** between the chains. The green, gray-40%, red, blue, and gray-25% colors represented $Mn^{2+}$, C, O, N, and H atoms, respectively.

probably due to the antiferromagnetic couplings between $Mn^{2+}$ ions and the photogenerated radicals. Unlike the as-prepared samples, **2** showed an asymmetric ESR signal at $g = 2.004$ (Fig. 2f) after light irradiation, confirming the photogenerated stable radicals. Furthermore, the spin concentration of **2a** was measured from the ESR spectra to estimate the amount of photogenerated radicals, and a spin concentration of $\sim 3.4 \times 10^{19}$ spins $mol^{-1}$ was obtained after irradiating the powder samples for 180 min by the 250-W Xe-lamp at ambient conditions. Compared with the theoretical value ($\sim 1.2 \times 10^{24}$ spins $mol^{-1}$) for **2**, the obtained spin concentration of **2a** is clearly smaller. This discrepancy is because photochromic reactions mainly occur at material surfaces. Owing to the similarity in the similar photochromic behavior, the spin concentration of **1a** should also be smaller than the theoretical value[50,51]. Thus, crystal samples of both compounds were ground to powder, and the light-irradiation time for photochromic experiments was increased to at least 180 min to ensure the sufficiently photogenerated radicals. Magnetic susceptibility measurements of **2a** revealed the formation of radicals (Supplementary Fig. 17).

To confirm the origin of the photogenerated radicals, the photochromic behavior of the $H_2ADC$ ligand was examined under the same conditions. $H_2ADC$ shows color changes from yellow to deep yellow after Xe-lamp illumination (Supplementary Fig. 18). The changes in the time-dependent UV–Vis and emission spectra also suggest the formation of $H_2ADC^{\bullet}$ radicals upon light irradiation (Supplementary Figs. 19 and 20). The ESR spectrum of $H_2ADC$ shows a signal with $g = 2.004$, demonstrating the formations of radical species after illumination (Supplementary Fig. 21), which is also confirmed by the unchanged IR spectra before and after illumination (Supplementary Fig. 22). Compared with the traditional electron transfer photochromic structures (i.e., the electron transfer process occurring in the separated donor and acceptor ligands)[52–54], the $H_2ADC$ molecule herein showed self-photochromic properties with intraligand or/ and interligand electron transfer processes. The photochromic behavior for both compounds is therefore originated from the photogeneration of $ADC^{\bullet}$ radicals. Furthermore, when both

$H_2ADC$ and **1** are dissolved in DMF and water solvent, respectively, photochromism is still observed in solutions with obvious variations in color and liquid-state UV–Vis spectra (Supplementary Figs. 23 and 24). This result suggests that intramolecular electron transfer is dominant in the photochromic behavior. Compared with $H_2ADC$, photochromism in **1** and **2** is more obvious, as indicated by more significant changes in both the color and spectra, which also indicates a metal-assisted electron transfer process. Since photogenerated radicals delocalize in large $\pi$-conjugated anthracene components, the lifetime of **1a** was estimated from the ESR spectra, which could maintain the charge-separated state for at least 3 months after photochromism, similar to other photochromic materials with stable radicals[37].

To explore the electron transfer path of photogenerated radicals in **1**, single crystals of **1** and **1a** were firstly analyzed utilizing the same single crystal. After light irradiation, the volume slightly shrinks with a decrease in H-bonding interactions, similar to other reported electron transfer photochromic complexes[55]. The transfer pathway for **1** was confirmed by density functional theory (DFT) calculations, and a method with the basis set B3LYP/6-311G(d) using the Gaussian 09 program was performed to calculate the spatial distributions of the highest occupied molecular orbital (HOMO) and lowest unoccupied molecular orbital (LUMO) levels. As shown in Fig. 2g, h, and Supplementary Fig. 25, electrons are mainly distributed in the ADC ligands, indicating that electron-transfer photochromism in **1** is mainly originated from the photoactive $H_2ADC$ ligands. However, the electron distribution in the HOMO is dominantly located on the carboxyl groups, whereas that in LUMO is dominant on the anthracene rings. The localization of HOMO and LUMO suggests the HOMO–LUMO transition is attributed to intraligand electron transfer from carboxyl groups to anthracene motifs[56]. For light irradiated **1a**, DFT calculations also revealed that electron distribution is mainly located on the ADC ligands (Supplementary Fig. 26), and the photogenerated radicals are delocalized in the ADC units. Thus, the photogenerated $ADC^{\bullet}$ radical species coordinate with $Mn^{2+}$ centers to trigger strong magnetic couplings and induce photomagnetic

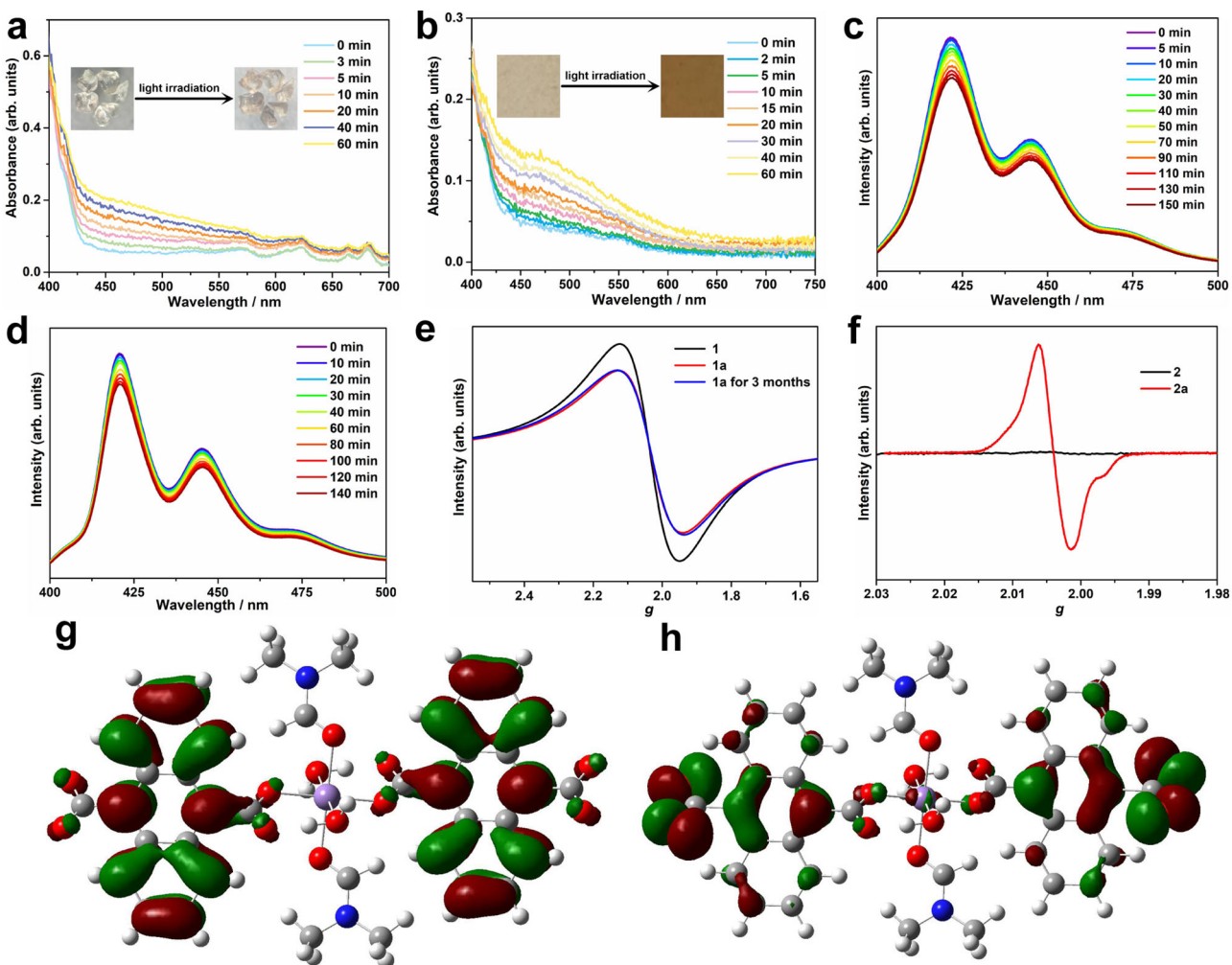

**Fig. 2 Characterization and mechanism of photochromism.** Time-dependent UV–Vis spectra of **1** (**a**) and **2** (**b**) upon irradiation. Insert: photos of color changes for crystal samples **1** and powder samples **2**; time-dependent photoluminescent spectra of **1** (**c**) and **2** (**d**) excited at 283 nm; room temperature ESR spectra of **1** (**e**) and **2** (**f**) under a frequency of 9.84 GHz before and after light irradiation; the calculated spatial distributions of HOMO (**g**) and LUMO (**h**) of **1** at the B3LYP/6-311G(d) level.

behavior[54]. Furthermore, the frontier molecular orbitals from HOMO − 4 to LUMO + 4 for **2–4** were also calculated, and electrons are distributed in the ADC ligands in all the compounds (Supplementary Figs. 27–29). However, the DFT results show that electron transfer occurs ont only in one ADC ligand, but also in the adjacent ADC ligands coordinated with the same metal center. The metal ions should also participate in the electron transfer process, because the sole $H_2ADC$ ligand exhibits self-photochromic phenomenon, and the constructed isostructural compounds show different photochromic behavior. This metal-assisted ligand-to-ligand electron transfer has been widely studied in many works[39,47].

**Photomagnetic properties**. Magnetic susceptibility of **1** was measured before and after light irradiation to explore the photomagnetic behavior. The $\chi T$ ($\chi$ is molar susceptibility and $T$ represents temperature) value for the as-prepared sample of **1** is 4.69 cm$^3$ K mol$^{-1}$ at room temperature (Fig. 3a), which is slightly higher than that of isolated high-spin (HS) Mn$^{2+}$ species ($S = 5/2$, 4.375 cm$^3$ K mol$^{-1}$) with $g = 2.0$. As the temperature decreases, $\chi T$ decreases slightly to 3.59 cm$^3$ K mol$^{-1}$ at 2 K. A Curie-Weiss plot gives a negative Weiss constant $\theta$ of −0.54 K (Fig. 3a, insert), indicating weak antiferromagnetic interactions. After irradiation,

$\chi T$ of **1a** at 300 K decreases to 4.34 cm$^3$ K mol$^{-1}$, showing a remarkable photodemagnetization effect after the generation of radicals, and antiferromagnetic coupling is observed, even at room temperature[42,49]. After experiencing a slight decrease, the curve shows a sudden drop with $T_{1/2\downarrow} = 62.7$ K. Then, the curve directly decreases to a lower value at 10 K. In contrast, the $\chi T$ curve is not followed but increased to the room-temperature value with another abrupt change ($T_{1/2\uparrow} = 239.8$ K), showing a thermal hysteresis loop with a large width of 177 K (Fig. 3b and Supplementary Fig. 30). As a result, we achieved the magnetic bistability with thermal hysteresis in the electron transfer photochromic materials.

To confirm this hysteresis, the observed thermal hysteresis was remeasured between 40 and 250 K, and the transition temperature and thermal hysteresis were nearly unchanged under different sweep rates (Fig. 3c), showing a stable and reproducible magnetically bistable property. Furthermore, direct-current magnetic measurements were conducted for **2a** with a diamagnetic Zn$^{2+}$ ion, **3** with an anisotropic Ni$^{2+}$ ion and **4** with an orbital angular-momentum contribution of a Co$^{2+}$ ion under the same experimental conditions. For **2a**, the photogenerated radicals show a characteristic of temperature-independent paramagnetism (Supplementary Fig. 17), whereas weak antiferromagnetic coupling without thermal hysteresis is observed for **3** and **4** (Supplementary Fig. 31), indicating that the

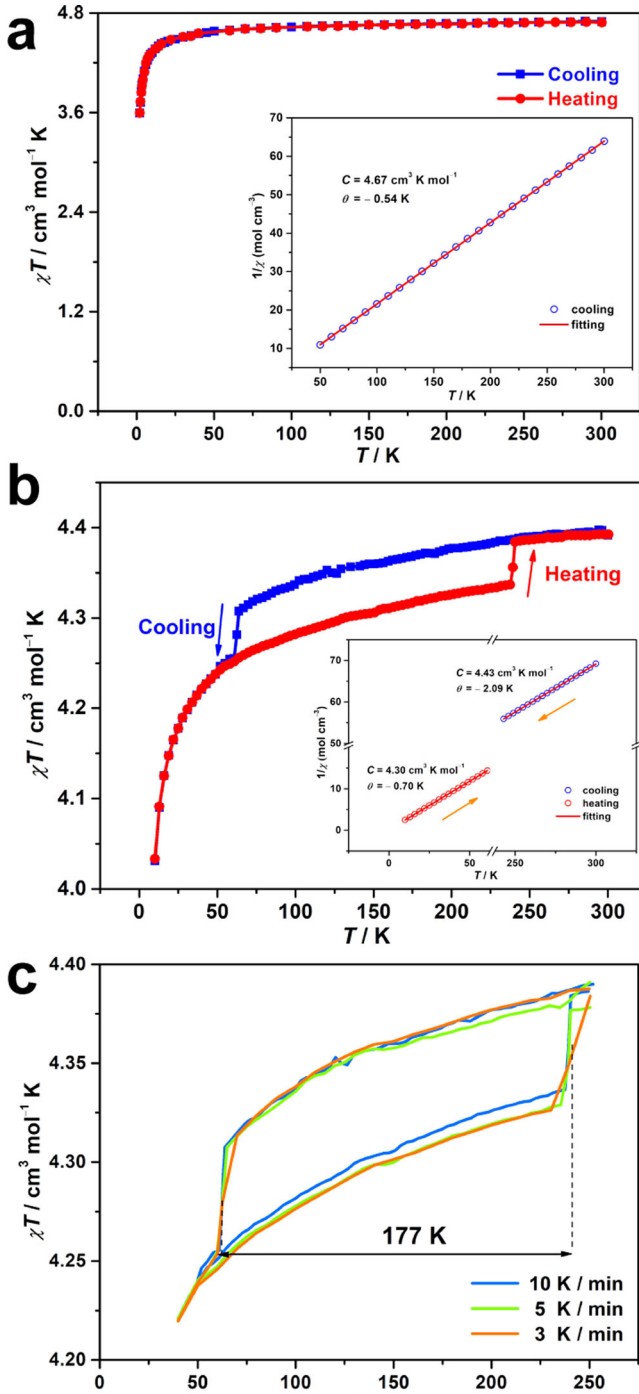

**Fig. 3 Magnetic susceptibility measurements before and after light irradiation.** Temperature dependence of $\chi T$ for **1** (**a**) and **1a** (**b**) in the cooling (blue) and heating (red) modes at a direct-current field of 1000 Oe. Insert: plots of $1/\chi$ versus $T$ for **1** and **1a** with Curie-Weiss fitting. $C$ and $\theta$ represent Curie temperature and Weiss constant; **c** temperature dependence of $\chi T$ for **1a** in the transition temperature regions with different sweep rates.

thermal hysteresis in **1a** is originated from the variations in magnetic coupling between the photogenerated radicals and $Mn^{2+}$ ions. In general, large thermal hysteresis loops in materials are very important for practical applications, while the number of complexes with hysteresis larger than 100 K is still limited (Supplementary Table 16). Compound **1a** showed remarkably wide and reproducible

thermal hysteresis loops (177 K) among the reported magnetically bistable molecular systems[20–24]. The magnetic susceptibility data for **1a** follows the Curie-Weiss trend (Fig. 3b, insert) with the fitting parameters of Curie temperature $C = 4.43$ cm$^3$ K mol$^{-1}$ and $\theta = -2.09$ K, and $C = 4.30$ cm$^3$ K mol$^{-1}$ and $\theta = -0.70$ K in the high- and low-temperature regions, respectively. The difference in the $\theta$ values at the high- and low-temperature regions for **1a** is due to the variations in magnetic coupling originating from the structural changes.

Furthermore, the exchange coupling constant $J$ between the photogenerated ADC$^\bullet$ radicals and $Mn^{2+}$ ions was obtained by DFT calculations to confirm the magnetic couplings. Zhang et al.[57] used the popular hybrid functional O3LYP[58] to obtain the closest exchange coupling constants between $Co^{2+}$ and ADC$^\bullet$ radicals. To obtain the exchange coupling constant $J$ between $Mn^{2+}$ and radicals, we first extracted a fragment of ADC$^\bullet$–$Mn^{2+}$–ADC$^\bullet$ (Fig. 4a) from the 1D chain of **1a** and then used O3LYP with ORCA 5.0.2[59] to calculate them. Def2-TZVP with auxiliary coulomb fitting basis set of SARC/J[60] was used for all atoms, and the zero-order regular approximation was employed for the scalar relativistic effect in the calculation. The tight grid of DEFGRID3 and tight convergence criteria were selected to ensure that the results converged well with respect to the technical parameters. For simplification, only the nearest neighboring $Mn^{2+}$–ADC$^\bullet$ exchange interactions are considered. First, we calculated the energy of the HS state ($S_{HS} = 1/2 + 1/2 + 5/2 = 7/2$) and then flipped the spins on all the atoms except for $Mn^{2+}$ to obtain the energy of the low-spin (LS) state ($S_{LS} = -1/2 - 1/2 + 5/2 = 3/2$). The spin-density distributions (Fig. 4b, c) show that the spin densities in the HS and LS states are mainly distributed over $Mn^{2+}$ and only one ADC$^\bullet$ because the nearest $Mn^{2+}(H_2O)_2(DMF)_2$ around the other ADC$^\bullet$ is omitted. The corresponding spins on $Mn^{2+}$ and one ADC$^\bullet$ are 4.896 and 2.019 in the HS state and 4.889 and $-1.970$ in the LS state, respectively. Thus, only the exchange coupling between $Mn^{2+}$ and one ADC$^\bullet$ should be considered for the created fragment. The spins of $Mn^{2+}$ and one ADC$^\bullet$ are 5/2 and 1, respectively. According to Hamiltonian, $\hat{H} = -2J_{Mn-ADC}\hat{S}_{Mn}\hat{S}_{ADC}$, the $Mn^{2+}$–ADC$^\bullet$ coupling constant $J_{Mn-ADC}$ was obtained based on Eqs. 1 and 2 using the spin- and non-spin-projected approaches, respectively[61–63].

$$J_{Mn-ADC} = \frac{E_{LS} - E_{HS}}{10} \text{(spin projection)} \quad (1)$$

$$J_{Mn-ADC} = \frac{E_{LS} - E_{HS}}{12} \text{(non − spin projection)} \quad (2)$$

$J_{Mn-ADC}$ calculated using Eqs. 1 and 2 are $-11.4$ and $-9.5$ cm$^{-1}$, respectively. The negative $J_{Mn-ADC}$ values suggest exchange couplings between $Mn^{2+}$ and ADC$^\bullet$ are antiferromagnetic for the 1D chain of **1a**, constant with the Curie-Weiss fitting results.

## Discussion
Magnetic bistablity has been reported for the molecular systems of spin crossover[6–8], metal-to-metal/ligand electron transfer[9–13], organic radicals with intra- or intermolecular electron-exchange interactions[14–16,64], spin-Peierls-type transition[65], transition metal complexes with dynamic coordination environments[17–19,66,67], and angular-momentum quenching of $Co^{2+}$ complexes[68–70]. However, the light-induced thermal hysteresis of **1** originates from the variations of magnetic couplings and cooperativity, which is distinct from the reported examples. To further explore the magnetic behavior of **1a**, variable-temperature ESR spectra were conducted between 300 and 100 K for **1**, **1a**, and **2a**. As shown in Fig. 5a, the ESR spectra profile of **1a** remain unchanged, while the intensity gradually increases upon cooling, similar to the ESR variation for **1** before irradiation (Supplementary Fig. 32). For **2a** in Fig. 5b, the intensity of the radical signal decreases sharply as

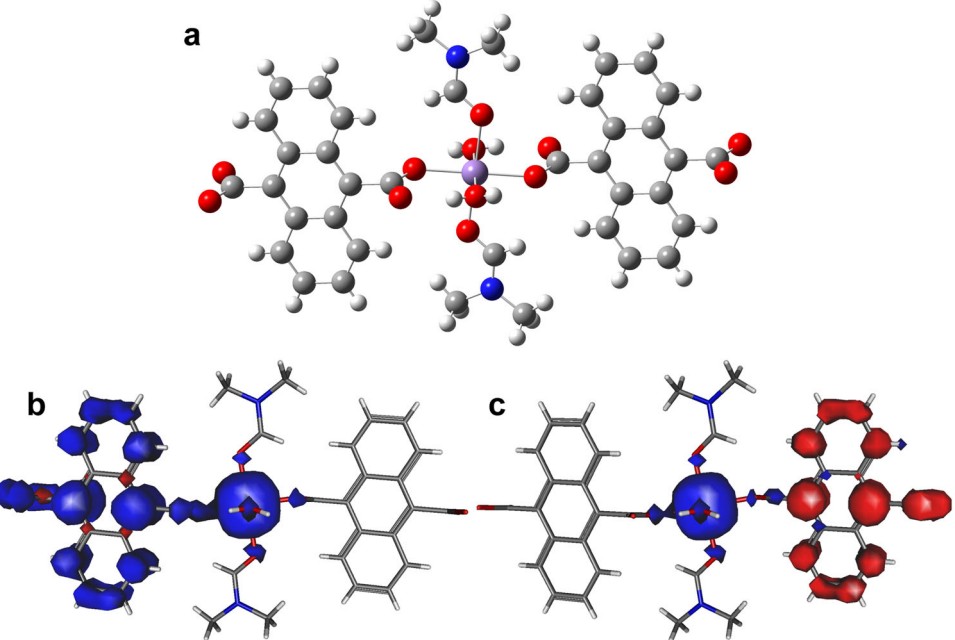

**Fig. 4 DFT calculations of magnetic couplings. a** Extracted fragment of ADC•–Mn²⁺–ADC• from compound **1a**. H atoms are omitted for clarity; Spin density distributions for the extracted fragment in the HS (**b**) and LS (**c**) states (blue and red regions indicate positive and negative spin populations, respectively, and the isodensity surface corresponds to a value of $0.002\,e^- \text{bohr}^{-3}$).

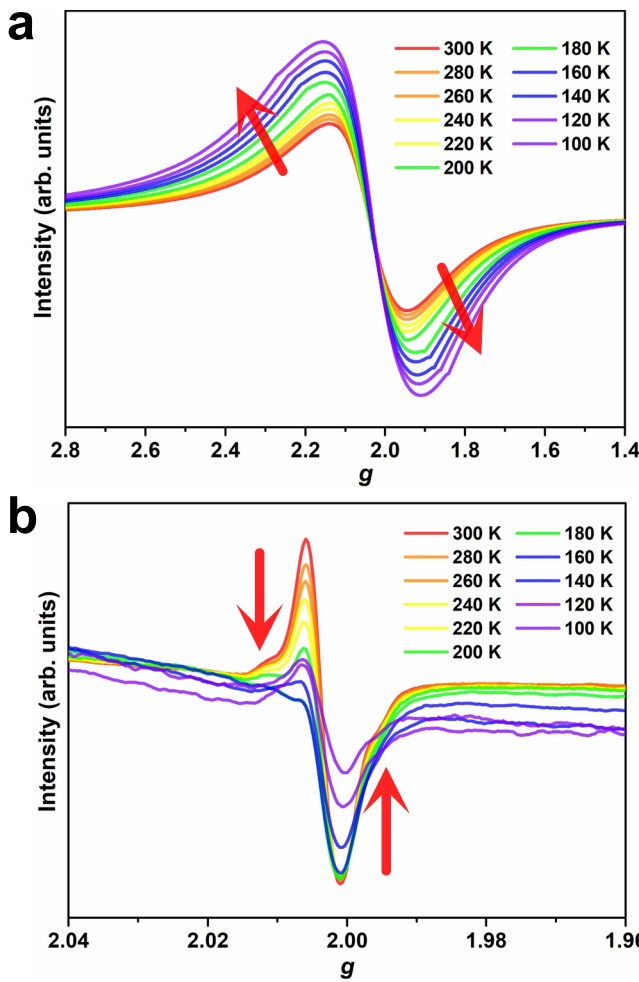

**Fig. 5 ESR spectra.** Variable-temperature ESR spectra for **1a** (**a**) and **2a** (**b**) under a frequency of 9.41 GHz at solid-state.

the temperature decreased to 100 K, suggesting that the singlet diradicals appeared due to antiferromagnetic couplings[71,72]. Based on the variable-temperature ESR data and magnetic analyses of **2a**, **3**, and **4**, Mn²⁺ ions antiferromagnetically couple with diradicals, inducing spin topology changes from a single ion to a single chain, further resulting in the magnetically bistable state. Compared with the π–π stacking interactions actuating thermal hysteresis in spin transition complexes, the interchain H-bonding interactions in **1a** connect the photogenerated radical-Mn²⁺ single chains to an extended 2D structure, and the enhanced cooperativity may also contribute to the large hysteresis, similar to that of other SCO complexes[2,30].

Single-crystals of **1a** were characterized at different temperature regions to further understand the hysteresis. The Mn–O bond lengths and angles remain unchanged upon cooling, indicating no spin transition in the metal centers[73,74]. However, the ADC ligand with a monodentate coordination mode rotates along the Mn–O2 bond as the temperature decreases, forming a dihedral angle of 2.724(2)° between the rings at high and low temperatures (Fig. 6a). As a result, the Mn1–O2–C8 bond angle tilts from 130.96(148)° to 128.40(109)° upon cooling to 50 K, and the twist angle of the Mn1–O2–C8–C3 torsion angle changes to 163.29(146)°, 159.91(150)°, and 159.10(106)° at 300, 93, and 50 K, respectively. This decrease in the torsion angle upon cooling induced weaker magnetic couplings between photogenerated radicals and Mn²⁺ centers, which was confirmed by the magnetic behavior of **1a**. Furthermore, DMF and water molecules coordinated to the Mn centers varies with molecular orientation (Fig. 6b and Supplementary Fig. 33). The coordinated DMF and water molecules in the overlapped structures show large displacement with an $O4_{300\,K}$–Mn1–$O4_{50\,K}$ angle of 5.58° and an $O3_{300\,K}$–Mn1–$O3_{50\,K}$ angle of 7.44°, whereas the angles O2–Mn1–O3, O2–Mn1–O4, and O3–Mn1–O4 change from 91.27(8), 91.65(8) and 88.27(9) to 89.60(5)°, 92.33(4)° and 85.38(5)° upon cooling to 50 K, respectively. Thus, the geometrical configuration of the Mn²⁺ center suffers a more and abnormal distortion in the low temperatures (CshM = 0.050 at 300 K; CshM = 0.136 at 93 K; CshM = 0.149 at 50 K,

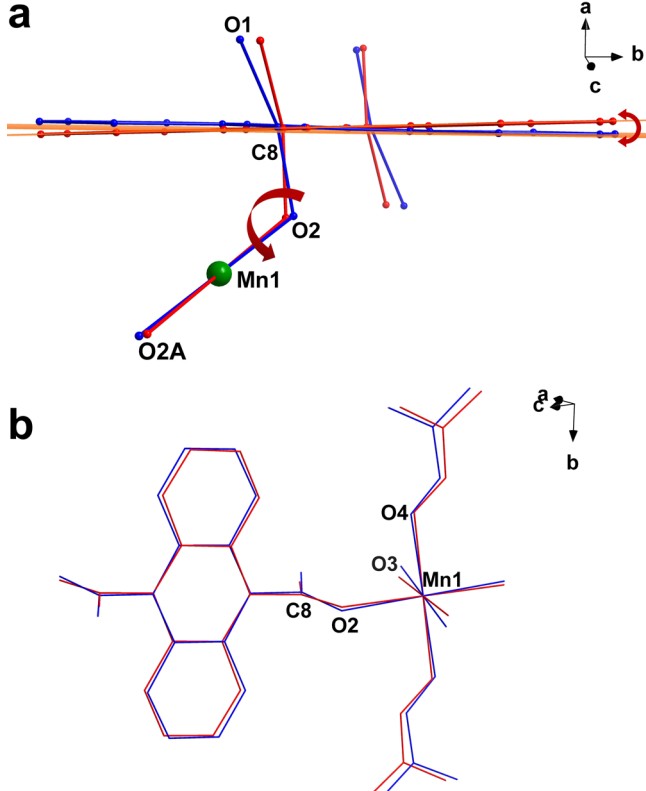

**Fig. 6 Variations of crystal structures.** Molecular structure overlap of **1a** at 50 K (blue) and 300 K (red) at different directions (**a**, **b**). Anthracene ring rotates along Mn–O2 bond and forms a dihedral angle of 2.24° between the rings from 300 (red line) to 50 K (blue line), suggesting the rotation of ADC units. H atoms, parts of DMF and water molecules are omitted for clarity.

Supplementary Table 10). The rotation of ADC alters the Mn1−O2−C8 angle, leading to structural changes in the Mn²⁺ center, inducing variations in magnetic interactions between Mn²⁺ ions and photo-triggered radicals, which resulted in pre-eminent magnetic bistability with the wide thermal hysteresis.

In summary, we synthesized a series of transition metal complexes using anthracene-derivative ligand as a photoactive component. Structural, spectroscopic, DFT calculations and magnetic characterizations revealed that the Mn²⁺ and Zn²⁺ analogs exhibited electron-transfer photochromic behavior. For Mn²⁺ congener, accompanied by the radical-actuated coloration, the spin topology changed from a single ion to a single chain, inducing an obviously photomagnetic response with wide thermal hysteresis after light irradiation. Compared with the non-photochromic isomorphism of Ni²⁺ compound with anisotropy and Co²⁺ complex with large orbital angular-momentum contribution, in which no photochromism and thermal hysteresis were observed for both compounds after light irradiation, this hysteresis anomaly is attributed to the variations in magnetic couplings between Mn²⁺ ions and photogenerated diradicals. The structural modifications between high and low temperatures are originated from changes in the Mn1−O2−C8 angle due to the rotation of ADC units, inducing variations in magnetic couplings between Mn²⁺ ions and photogenerated radicals, further resulting in the remarkably wide 177 K thermal hysteresis. Unprecedented magnetic bistability was achieved with an large thermal hysteresis loop in radical-actuated photochromic materials, providing a class of photochromic materials for the rational design of

molecular magnets with large thermal hysteresis and promoting the development of the interdisciplinary field of molecular magnetism, electronics, and photonics.

## Methods

**Materials**. All chemicals were purchased commercially. The organic H2ADC ligand was recrystallized in methanol for photochromism and further syntheses.

**Synthesis of [Mn(ADC)(H₂O)₂(DMF)₂]ₙ (1)**. A mixture of MnCl₂·4H₂O (0.20 mmol, 0.039 g), H₂ADC (0.050 mmol, 0.013 g), DMF (1.0 mL), and H₂O (4.0 mL) were sealed in a glass vial and heated to 90 °C for 5 days. Colorless X-ray-quality crystals were formed and dried in air. Yield: 0.049 g (49%) based on MnCl₂·4H₂O. Compound **1a** was obtained by 250-W Xe-lamp irradiation of **1** at room temperature for 180 min. Elemental analysis (%): calcd for **1** (%): calcd for C₂₂H₂₆N₂MnO₈ (501.39): C, 52.70; H, 5.23; N, 5.59. Found: C, 52.62; H, 5.11; N, 5.54. IR (KBr pellet, cm⁻¹): 3515(w), 2921(w), 1656(s), 1569(m), 1434(m), 1328(w), 1276(w), 1099(w), 1028(w), 861(w), 790(m), 671(m), and 467(w). For **1a** (%): calcd for C₂₂H₂₆N₂MnO₈ (501.39): C, 52.70; H, 5.23; N, 5.59. Found: C, 52.57; H, 5.31; N, 5.66.

**Synthesis of [Zn(ADC)(H₂O)₂(DMF)₂]ₙ (2)**. The crystals were prepared in a similar way with compound **1** by using Zn(NO₃)₂·6H₂O (0.20 mmol, 0.059 g). Yield: 0.045 g (44%) based on Zn(NO₃)₂·6H₂O. Compound **2a** was obtained by 250-W Xe-lamp irradiation of **2** at room temperature for 180 min. Elemental analysis (%): calcd for **2** (%): calcd for C₂₂H₂₆N₂ZnO₈ (511.82): C, 51.62; H, 5.12; N, 5.47. Found: C, 51.72; H, 4.95; N, 5.54. IR (KBr pellet, cm⁻¹): 3567(w), 3436(w), 2931(s), 1654(s), 1554(s), 1488(w), 1432(m), 1374(w), 1322(m), 1276(w), 1245(w), 1099(m), 861(w), 782(m), 676(m), 479(w). For **2a** (%): calcd for C₂₂H₂₆N₂ZnO₈ (511.82): C, 51.62; H, 5.12; N, 5.47. Found: C, 51.81; H, 5.03; N, 5.43.

**Synthesis of [Ni(ADC)(H₂O)₂(DMF)₂]ₙ (3)**. The crystals were prepared in a similar way with compound **1** by using Ni(NO₃)₂·6H₂O (0.20 mmol, 0.058 g). Yield: 0.048 g (47%) based on Ni(NO₃)₂·6H₂O. Elemental analysis (%): calcd for C₂₂H₂₆N₂NiO₈ (505.16): C, 52.31; H, 5.19; N, 5.55. Found: C, 52.32; H, 5.08; N, 5.34. IR (KBr pellet, cm⁻¹): 3598(w), 3430(m), 2976(m), 2929(s), 1646(s), 1550(m), 1436(m), 1374(w), 1326(w), 1272(w), 1101(w), 1051(w), 871(w), 782(m), 678(m), 584(w) 484(w).

**Synthesis of [Co(ADC)(H₂O)₂(DMF)₂]ₙ (4)**. The crystals were prepared in a similar way with compound **1** by using Co(NO₃)₂·6H₂O (0.20 mmol, 0.058 g). Yield: 0.040 g (40%) based on Co(NO₃)₂·6H₂O. Elemental analysis (%): calcd for C₂₂H₂₆N₂CoO₈ (508.38): C, 52.28; H, 5.19; N, 5.54. Found: C, 52.37; H, 5.28; N, 5.44. IR (KBr pellet, cm⁻¹): 3561(w), 2933(s), 1644(s), 1552(m), 1439(m), 1376(w), 1318(m), 1276(w), 1099(w), 780(m), 671(m), 605(w), and 479(w).

**Physical property measurements**. Elemental analyses were performed on a PerkinElmer 240C analyzer. The UV–Vis spectra were applied on a Puxi Tu-1901 spectrophotometer. The luminescence curves were recorded on a Hitachi F-7000 Fluorescence spectrometer. IR curves were collected on an ABB Bomen MB 102 series FT−IR spectrometer. The ESR spectra were measured on CIQTEK EPR200-Plus with a continues-wave X band frequency of 9.84 GHz for H₂ADC ligand, and a Bruker E500 spectrometer with continues-wave X band frequencies of 9.84 GHz at room temperature and 9.41 GHz at low temperatures for **1** and **2**. Magnetic measurements of the polycrystalline samples were carried out on a Quantum Design SQUID PPMS magnetometer. PXRD curves were performed on a Rigaku diffractometer with a Cu-target tube and a graphite monochromator. Simulation of the PXRD curves were performed by the single-crystal data and diffraction-crystal module of the Mercury (Hg) program available free of charge via the Internet at http://www.iucr.org. Furthermore, a Rietveld refinement of PXRD between the experimental pattern and the single crystal data were also performed. For the light irradiation experiments, a Perfect Light PLS-SXE 300 Xe-lamp (320–780 nm, 250 W, at least 180 min) was equipped to prepare the colored samples of **1** and **2** for elemental analyses, IR, crystal XRD, UV–Vis, PXRD, ESR and magnetic studies.

**Computational methods**. Molecular orbital calculations were performed using the Gaussian 09 program and the basis set B3LYP/6-311G(d) method and adapted from the crystal X-ray data. The exchange coupling constant *J* between photo-generated radicals and Mn²⁺ ions was calculated by DFT calculations using the popular hybrid functional O3LYP with ORCA 5.0.2.

**Single-crystal X-ray crystallographic study**. The single-crystal X-ray diffraction data of **1a** with different temperatures were collected on a Bruker D8 Venture CMOS-based single-crystal X-ray diffractometer equipped with a graphite-monochromated Mo-Kα radiation (λ = 0.71073 Å) using the SMART and SAINT programs. **1–4** were collected on a Rigaku SCX-mini diffractometer with Mo-Kα

radiation ($\lambda = 0.71073$ Å) at room temperature. The SHELX-2016 software was used to solve all those structures[75].

## Data availability

Detailed crystallographic data for **1–4** were summarized in Supplementary Tables 1 and 2, and the selected bond lengths and angles were listed in Supplementary Tables 3–9. The X-ray crystallographic data for the structure reported in this article has been deposited at the Cambridge Crystallographic Data Centre (CCDC) with the number of CCDC 2092423 for **1**, 2092424 for **1a** at 300 K, 2092425 for **1a** at 93 K, 2092426 for **1a** at 50 K, 2092427 for **2**, 2092428 for **3**, and 2092429 for **4**. These data can be obtained free of charge from The Cambridge Crystallographic Data Center via www.ccdc.cam.ac.uk/data_request/cif. We declare that the main data supporting the findings of this study are available within the article and its supplementary information files. Source data are provided with this paper. All relevant source data are also available from the corresponding author upon request.

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

## Acknowledgements
This research was supported by the National Natural Science Foundation of China 22071126 (G.M.W.), 21901133 (J.X.H.), 22171155 (J.X.H.), 91961114 (T.L.), 22025101 (T.L.) and China National Postdoctoral Program for Innovative Talents BX20180147 (J.X.H.). The authors wish to acknowledge Professor Hiroki Oshio (Dalian University of Technology, China) for his improvement in language and helpful discussions regarding the experimental design.

## Author contributions
J.X.H. and G.M.W. conceived the idea and designed the experiments. Q.L. did the synthesis, IR, photoluminescence, UV–Vis, PXRD, ESR, elemental analysis, and solved the crystal structures. J.X.H. and H.L.Z. measured the magnetic data. J.X.H. performed the DFT calculations. J.X.H., G.M.W., and T.L. analyzed the data. Z.N.G. and Q.Z. assisted Q.L. in testing photochromic measurements. J.X.H., G.M.W., and T.L. wrote the manuscript. J.X.H. and G.M.W. in Qingdao and T.L. in Dalian conceived and supervised the project. All authors discussed the results and commented on the manuscript.

## Competing interests
The authors declare no competing interests.
