## [Peer Review File · Nature Communications]

Achieving Large Thermal Hysteresis of an Anthracene-Based Manganese(II) Complex via Photo-Induced Electron TransferREVIEWER COMMENTS

Reviewer #1 (Remarks to the Author):

Hu and co-workers describe the synthesis and characterisation of a series of anthracene complexes, with particular focus on the magnetism of the manganese(II) complex which demonstrates an impressively wide thermal hysteresis of the magnetic properties.

There are some issues with the language and another proof is necessary to improve the clarity of the authors message.

A major concern with the body of work as presented is the lack of thorough DFT/computational analysis. Of the 4 complexes reported, only complex 1 is subjected to DFT calculations and that is only on the HOMO/LUMO of the molecule. The abstract and conclusion state that both 1 AND 2 are subjected to DFT but I am unable to see any calculations for complex 2.

Main text;

Line 8 - "colossal" undefined, ambiguity is understandable in the title however a bit more clarity here would be appreciated. Especially to contextualise the 177 K mentioned later in the abstract.

Line 9 - Molecular formula for chain complex lacks definition of number of repeat units (x or n etc).

Also, notation not consistent with line 54

Line 28-30 - poorly structured sentence which could be improved.

Line 32 - "indispensable" does not seem to be the correct word for this sentence. Also, "only a few complexes" would be improved by a number to show understanding of the field.

Line 39 - Still no definition of "wide thermal hysteresis".

Line 43 - Wavelength of irradiation used would be good information to include.

Line 44/45 - Too many adjectives.

Line 46 - "Magnets" should be "magnetism".

Line 49 - "substantially" is unnecessary in this sentence.

Line 58 - T1/2 arrows not defined.

Line 61 - Reference missing.

Experimental section;

Yields only as percentages, no mass yields.

Section describing equipment used should go before description of synthesis.

Line 100 - *via* is in italics but this is not consistently applied throughout manuscript.

Line 105/107 - inconsistent notation of X-ray wavelength used.

Line 112 - "was" should be "is".

Line 149 - "totally" should be replaced with "could in total"

Line 151 - "should" needs explaining and/or a reference.

Line 154 - "ensure the amount..." sentence needs ending, is it to ensure the amount is increased/consumed etc?

Line 191 - "may result in.." needs either evidence or a reference to back up this claim.

Line 194 - Value of isolated high spin Mn²⁺ would be beneficial

Line 211 - "orbital angular momentum contribution"

Line 240 - "write formula" presumably is supposed to be a reference complex?

Line 249, 250, 251 - there are no ESD values compared with later in the manuscript.

Conclusion - DFT for complex 2 referred to again but no DFT for 2 in manuscript.

Figure 1 - I think the use of capped sticks is limited here, ellipsoids would be better &/or inclusion of a chemdraw would help. Caption notes green is M²⁺ and also used to shade the M octahedra, doesn't help with clarity of image.

SI;

Figure S1 is described as "The molecular structure for all the compounds" whereas the main text refers to figure S1 as the structure of complex 1.

Figure S9 states "The color changes of compound 3 and 4" but it is very difficult to discern any colour changes in either a) or b).

Figure S10 has IR for both complex 3 & 4, but not explicitly clear which is which, compared to Figure S11 where the authors have noted in the caption which UV-Vis spectrum belongs to which.

A greater degree of consistency in the manuscript and SI would be appreciated.

Whilst the work has clearly been conducted to a rigorous standard including a good deal of data collection, I would not recommend that this manuscript be published in Nat. Commun. for the reasons described. Instead, I would suggest the authors submit to a different journal.

Reviewer #2 (Remarks to the Author):

I think this is a nice paper of valid scientific results, however I have small issues regarding the crystallographic part:

- Every single structure shows the same ADDSYM alert, sometimes at B level and sometimes at C level, regarding the possible wrong space group choice. I would like to see the authors dealing with this at least with a comment.

- Comparing powder pattern of different compounds simply "by eye" with a calculated one from Mercury is not the best way of doing. Instead, I strongly recommend a Rietveld refinement between the experimental pattern and the one from the single crystal data.

Once these issues have been dealt with I recommend this manuscript for publication.

Reviewer #3 (Remarks to the Author):

This work for the first time achieved the magnetic thermal hysteresis loop with the temperature width of 177 K in the radical-actuated photochromic materials. It is very interesting for the readers of Nature Commun. I recommend it to be published in Nature Commun. after considering the following questions.

1) They indicate there are strong magnetic couplings between the photogenerated radicals and Mn²⁺. However, I don't find the exchange coupling constant J between them. I suggest authors to calculate J using DFT. The computational details can be found in Inorg. Chem. 2021,60,1007.

2) They conclude the thermal hysteresis of 1a was due to the variations of magnetic couplings between Mn²⁺ and photogenerated diradicals and interchain H-bonding interactions. 3a and 4a also have the photogenerated radicals, but why they have no thermal hysteresis. Please give more detail analyses.

Reviewer #4 (Remarks to the Author):

The paper reports a very important result on the obtaining of an electron transfer compound with an exceptionally large thermal hysteresis loop. Several compounds have been prepared and only one exhibits this remarkable behavior. Each compound has been deeply characterized to evidence and explain the origin of such large bistability. Even if the characterizations are mainly convincing, some information are lacking. I would have 3 major points to express before allowing this paper to be published in Nature Comm.

1. If the electron transfer is between a coordinated DMF and the anthracene units why compound 2 that shows the ET does not exhibit any hysteresis? The explanation concerning the absence of this ET in 3 and 4 focused on the absorption bands (page 78, line 143) seems also incomplete. What is the ET pathway? Through the metal center? Through space? What the impact of the metal on the redox potential that can favor or prevent the ET to occur?

2. I would strongly suggest to perform and show the DFT calculations for all the compounds to compare the ET pathway and efficiency. If the metal is not involved, it should not depend on its nature. Moreover, what are the LUMO and HOMO features in the photo-excited states? Since the XRD data are available, it might be interesting to perform the same calculations. This would help to understand the magnetic properties.

3. Concerning the magnetic properties and cooperativity, it would be interesting to use the DFT calculation to access the coupling constant between the radical and the metal, and compare it with data fitting. Moreover, it is not clear why in 1a the H bonding network leads to such large hysteresis. Is cooperativity really bigger, and the only explanation? Again why 2a does not show any thermally-induced ET?

Finally, I think the English might be improved and refs 6 and 7 changed to more general ones related to SCO like the Gutlich and Halcrow books.

Response to Reviewer Comments

Reviewer: 1

Hu and co-workers describe the synthesis and characterization of a series of anthracene complexes, with particular focus on the magnetism of the manganese(II) complex which demonstrates an impressively wide thermal hysteresis of the magnetic properties. There are some issues with the language and another proof is necessary to improve the clarity of the author's message.

Response: We are grateful for the kind and valuable comments from reviewer and the suggestions have enabled us to much improve our work. Here is our point-by-point response to the comments raised by the reviewer. Thanks very much for reviewing and improving our work.

Q1. A major concern with the body of work as presented is the lack of thorough DFT/computational analysis. Of the 4 complexes reported, only complex **1** is subjected to DFT calculations and that is only on the HOMO/LUMO of the molecule. The abstract and conclusion state that both **1** AND **2** are subjected to DFT but I am unable to see any calculations for complex **2**.

A. Thanks very much for the valuable suggestions from reviewer. The most important highlight in our work is the first realization of magnetic bistability with a large thermal hysteresis of 177 K in the electron transfer photochromic materials. However, as the reviewer suggested, the mechanism of the electron transfer process is vital and the thorough DFT/computational analysis for all the compounds is indispensable for improving our work. We also modified the description of the abstract, text and conclusion. For reliably explain the electron transfer process, we changed the structural model and extracted a fragment of ADC^{2-} - Mn^{2+} - ADC^{2-} from 1D chain and use the Gaussian 09 program and the basis set B3LYP/6-311G(d) method to re-calculate all these series of compounds. The DFT calculations indicated that the electron transfer process is mainly occurred in the ADC^{2-} ligand, and the localization of all the HOMO and LUMO suggests the HOMO-LUMO transition with an intraligand electron transfer from carboxyl groups to anthracene motifs. After light irradiation, the Mn^{2+} ions coordinated with the photogenerated ADC^{\cdot} radical analogs, further inducing the photomagnetic behavior with magnetic bistability in a large temperature region. We also

modified the manuscript as follows and added the DFT calculations in the supporting information. “The transfer pathway for **1** was confirmed by density functional theory (DFT) calculations, and the spatial distributions of the highest occupied molecular orbital (HOMO) and lowest unoccupied molecular orbital (LUMO) were calculated using the Gaussian 09 program and the basis set B3LYP/6-311G(d) method. As shown in Figures 2g, 2h, and S25, electrons were mainly distributed in the ADC²⁻ ligands, indicating that electron-transfer photochromism in **1** is mainly originated from the photoactive H₂ADC ligands. However, the electron distribution in the HOMO was dominantly located on the carboxyl groups, whereas that in LUMO was dominant on the anthracene rings. The localization of HOMO and LUMO suggests the HOMO–LUMO transition is attributed to intraligand electron transfer from carboxyl groups to anthracene motifs.⁵⁶ For light irradiated **1a**, DFT calculations also revealed that electron distribution is mainly located on the ADC ligands (Figure S26), and the photogenerated radicals are delocalized in the ADC units. Thus, the photogenerated ADC[•] radical species coordinate with Mn²⁺ centers to trigger strong magnetic couplings and induce photomagnetic behavior.⁵⁴ Furthermore, the difference in the frontier molecular orbitals from HOMO–4 to LUMO+4 for **2–4** was calculated, and electrons were distributed in the ADC²⁻ ligands in all the compounds (Figures S27–S29). However, the DFT result shows that electron transfer occurs not only in one ADC ligand, but also in the adjacent ADC ligands coordinate with the same metal center, because the sole ADC ligand exhibits self-photochromic behavior, and the constructed isostructural compounds show different photochromic behavior. The metal ions should also participate in the electron transfer process. This metal-assisted ligand-to-ligand electron transfer has been widely studied in many works.^{39,47.”}

Figure S25. HOMO, HOMO-1, HOMO-2, HOMO-3, HOMO-4, LUMO, LUMO+1, LUMO+2, LUMO+3 and LUMO+4 frontier molecular orbitals of compound **1** obtained by DFT calculations.

Figure S26. HOMO, HOMO-1, HOMO-2, HOMO-3, HOMO-4, LUMO, LUMO+1, LUMO+2, LUMO+3 and LUMO+4 frontier molecular orbitals of compound **1a** obtained by DFT calculations.

Figure S27. HOMO, HOMO-1, HOMO-2, HOMO-3, HOMO-4, LUMO, LUMO+1, LUMO+2, LUMO+3 and LUMO+4 frontier molecular orbitals of compound **2** obtained by DFT calculations.

Figure S28. HOMO, HOMO-1, HOMO-2, HOMO-3, HOMO-4, LUMO, LUMO+1, LUMO+2, LUMO+3 and LUMO+4 frontier molecular orbitals of compound **3** obtained by DFT calculations.

Figure S29. HOMO, HOMO-1, HOMO-2, HOMO-3, HOMO-4, LUMO, LUMO+1, LUMO+2, LUMO+3 and LUMO+4 frontier molecular orbitals of compound **4** obtained by DFT calculations.

Furthermore, we also calculated the exchange coupling constant J using DFT calculations to access the coupling constant between the photogenerated radicals and the Mn^{2+} ions. According to the reference *Inorg. Chem.* 2021, 60, 1007, Zhang *et al.*⁵⁷ had succeeded in using the popular hybrid functional O3LYP⁵⁸ to obtain the closest exchange coupling constants between Co^{II} and radicals to the experiment. To obtain the exchange coupling constants J between Mn^{2+} and radicals, we firstly extracted a fragment of ADC- Mn^{2+} -ADC (Figure 4a) from 1D chain of **1a**, and then used O3LYP with ORCA 5.0.2⁵⁹ to calculate them. Def2-TZVP with auxiliary coulomb fitting basis set of SARC/J⁶⁰ was used for all atoms, and zero order regular approximation was used for the scalar relativistic effect in the calculation. The tight grid of DEFGRID3 and tight convergence criteria were selected to ensure that the results are well converged with respect to technical parameters. For simplification, only the main nearest neighboring Mn^{2+} -ADC exchange interactions were considered.

We firstly calculated the energy of the high-spin (HS) state ($S_{\text{HS}} = 1/2 + 1/2 + 5/2 = 7/2$), and then flipped the spins on all of atoms except for Mn^{2+} to obtain the energy of the low-spin (LS) state ($S_{\text{LS}} = -1/2 - 1/2 + 5/2 = 3/2$). From the spin density distributions in Figure 4b and 4c, we find that the spin densities in the HS and LS states are mainly distributed over Mn^{2+} and only one ADC due to omitting the nearest $\text{Mn}^{2+}(\text{H}_2\text{O})_2(\text{DMF})_2$ around the other ADC.

Figure 4 (a) Extracted fragment of ADC–Mn²⁺–ADC from compound **1a**. H atoms are omitted for clarity; Spin density distributions for the extracted fragment in the HS (b) and LS (c) states (blue and red regions indicate positive and negative spin populations, respectively; the isodensity surface represented corresponds to a value of 0.002 e⁻ bohr⁻³).

The corresponding spin values on Mn²⁺ and one ADC are 4.896 and 2.019 in the HS state, and 4.889 and –1.970 in the LS state, respectively. Thus, we only need to consider the exchange coupling between Mn²⁺ and one ADC for our created fragment. The spins of Mn²⁺ and one ADC are 5/2 and 1, respectively. According to Hamiltonian: $\hat{H} = -2J_{Mn-ADC} \hat{S}_{Mn} \hat{S}_{ADC}$. The Mn²⁺–ADC coupling constant J_{Mn-ADC} was obtained through Eqs. 1 and 2 using the spin-projected and non-spin-projected approach, respectively.⁶¹⁻⁶³

$$J_{Mn-ADC} = \frac{E_{LS} - E_{HS}}{10} \quad (\text{spin projection}) \quad (1)$$

$$J_{Mn-ADC} = \frac{E_{LS} - E_{HS}}{12} \quad (\text{non-spin projection}) \quad (2)$$

The calculated J_{Mn-ADC} according to Eqs. 1 and 2 are –11.4 and –9.5 cm⁻¹, respectively. The negative J_{Mn-ADC} values suggest the exchange couplings between Mn²⁺ and ADC are antiferromagnetic for 1D chain of **1a**.

57. Lu, F., Li, J. X., Guo, W. X., Wang, B. L. & Zhang, Y. Q. Origin of Magnetic Relaxation Barriers in a Family of Cobalt(II)–Radical Single-Chain Magnets: Density Functional Theory and *Ab Initio* Calculations. *Inorg. Chem.* **60**, 1007–1015 (2021).

58. Handy, N. C. & Cohen, A. J. Left-Right Correlation Energy. *Mol. Phys.* **99**, 403–412 (2001).

59. Neese, F. ORCA—an *ab initio*, Density Functional and Semiempirical Program Package, version 5.0.2; Max-Planck institute for bioinorganic chemistry: Mülheim an der Ruhr, Germany. (2021).
60. Rolfes, J. D., Neese, F. & Pantazis, D. A. All-Electron Scalar Relativistic Basis Sets for the Elements Rb–Xe. *J. Comput. Chem.* **41**, 1842–1849 (2020).
61. Ruiz, E., Rodríguez-Forteza, A., Cano, J., Alvarez, S. & Alemany, P. About the Calculation of Exchange Coupling Constants in Polynuclear Transition Metal Complexes. *J. Comput. Chem.* **24**, 982–989 (2003).
62. Noodleman, L. & Case, D. A. Density-Functional Theory of Spin Polarization and Spin Coupling in Iron-Sulfur Clusters. *Adv. Inorg. Chem.* **38**, 423–470 (1992).
63. Bencini, A. & Totti, F. A Few Comments on the Application of Density Functional Theory to the Calculation of the Magnetic Structure of Oligo-Nuclear Transition Metal Clusters. *J. Chem. Theory Comput.* **5**, 144–154 (2009).

Main text;

Q2. Line 8 - "colossal" undefined, ambiguity is understandable in the title however a bit more clarity here would be appreciated. Especially to contextualise the 177 K mentioned later in the abstract.

A. We are grateful for the valuable suggestions from reviewer. As the reviewer suggested, we modified the word “colossal” as “large” in the title, abstract and the main text. We also instead this word as “extremely wide” in the main text.

Q3. Line 9 - Molecular formula for chain complex lacks definition of number of repeat units (x or n etc). Also, notation not consistent with line 54.

A. Thanks for the kind suggestions from the reviewer. As the reviewer suggested, we have modified the molecular formula for the chain complexes as $[M(ADC)(H_2O)_2(DMF)_2]_n$. Furthermore, the notation of molecular formula is consistent in the revised manuscript now.

Q4. Line 28-30 - poorly structured sentence which could be improved.

A. Thanks for your suggestions. We have improved and polished the English language and grammar in the revised manuscript.

Q5. Line 32 - "indispensable" does not seem to be the correct word for this sentence. Also, "only a few complexes" would be improved by a number to show understanding of the field.

A. We thank you for the reviewer’s valuable advises. We modified the word “indispensable”

as “vital” in the revised manuscript, and we also changed the sentence as “only five complexes have shown large and reproducible thermal hysteresis loops ($\Delta T > 100$ K).” Furthermore, the reported complexes with the thermal hysteresis loops ($\Delta T > 100$ K) are shown in Table S16.

Table S16. The complexes with the width of thermal hysteresis loops larger than 100 K.

Molecular formula	$T_{1/2\downarrow}$	$T_{1/2\uparrow}$	ΔT	reproducible	reference
Rb _{0.73} Mn[Fe(CN) ₆] _{0.91} ·1.4H ₂ O	147 K	262 K	116 K	√	20
[FeL ₂](BF ₄) ₂	350 K	495 K	145 K	√	21
Rb ^I _{0.64} Mn ^{II} ·[Fe ^{III} (CN) ₆] _{0.88} ·1.7H ₂ O	165 K	303 K	138 K	√	22
[Fe(bpp) ₂](CF ₃ SO ₃) ₂ ·H ₂ O	147 K	285 K	138 K	√	23
Rb _{0.94} Mn[Fe(CN) ₆] _{0.98} ·2.5H ₂ O	185 K	300 K	115 K	√	24
[FeL ₂](BF ₄) ₂ ·xH ₂ O	360 K	490 K	130 K	×	25
Co(py ₂ O)(3,6-DBQ) ₂	100 K	330 K	230 K	×	26
[Fe(qsal) ₂]NCSe·MeOH	212 K	352 K	140 K	×	27
[Fe(qsal) ₂]NCSe·CH ₂ Cl ₂	212 K	392 K	180 K	×	
[Fe(hyetrz) ₃](anion) ₂ ·3H ₂ O	100 K	370 K	270 K	×	28
[Fe(qsal-I) ₂]OTf·EtOH	139 K	252 K	113 K	×	R1
[Co(C12-terpy) ₂](BF ₄) ₂	258 K	□400 K	□142 K	×	R2
[Fe(qsal) ₂]NCSe·DMSO	209 K	324 K	115 K	×	R3
[Fe(qnal-OMe) ₂]BPh ₄ ·2MeOH	194 K	304 K	110 K	×	R4
[Fe(1-BPP-COOC ₂ H ₅) ₂](ClO ₄) ₂ ·CH ₃ CN	183 K	284 K	101 K	×	R5

References

- R1: Phonsri, W.; Harding, P.; Liu, L. J.; Telfer, S. G.; Keith S. M.; Moubaraki, B.; Ross, T. M.; Jameson, G. N. L.; Harding, D. J. Solvent modified spin crossover in an iron(III) complex: phase changes and an exceptionally wide hysteresis. *Chem. Sci.* **2017**, *8*, 3949–3959.
- R2: Han, Y.; Huynh, H. V. Pyrazolin-4-ylidenes: a new class of intriguing ligands. *Dalton Trans.* **2011**, *40*, 2141–2147.
- R3: Hayami, S.; Kawahara, T.; Juhász, G.; Kawamura, K.; Uehashi, K.; Sato, O.; Maeda, Y. Iron(III) spin transition compound with a large thermal hysteresis. *J. Radioanal. Nucl. Chem.* **2003**, *255*, 443–447.
- R4: Nakaya, M.; Shimayama, K.; Takami, K.; Hirata, K.; Alao, A. S.; Nakamura, M.; Lindoy, L. F.; Hayami, S. Spin-crossover and LIESST Effect for Iron(III) Complex Based on π - π Stacking by Coordination Programming. *Chem. Lett.* **2014**, *43*, 1058–1060.
- R5: Senthil Kumar, K.; Heinrich, B.; Vela, S.; Moreno-Pineda, E.; Bailly, C.; Ruben, M. Bi-stable spin-crossover characteristics of a highly distorted [Fe(1-BPP-COOC₂H₅)₂](ClO₄)₂·CH₃CN complex. *Dalton Trans.* **2019**, *48*, 3825–3830.

Q6. Line 39 - Still no definition of "wide thermal hysteresis".

A. Thanks for the advice from the reviewer. According to the reference (*J. Am. Chem. Soc.* 2001, 123, 11644–11650), a molecular compound that exhibits hysteresis can take two different electronic states between $T_{1/2\uparrow}$ and $T_{1/2\downarrow}$ depending on its history, and $T_{1/2\uparrow}$ and $T_{1/2\downarrow}$ are defined as the temperatures for which there are 50% high-spin and 50% low spin states in the warming and cooling modes, respectively. The width of thermal hysteresis (ΔT) is the difference between $T_{1/2\uparrow}$ and $T_{1/2\downarrow}$. Furthermore, the wide thermal hysteresis loops should be $\Delta T > 100$ K (*J. Am. Chem. Soc.* 2014, 136, 878–881). We also added the definition of "wide thermal hysteresis" in the revised manuscript.

Q7. Line 43 - Wavelength of irradiation used would be good information to include.

A. We thank you for the reviewer's valuable suggestion. The light used in the electron transfer photochromic materials is usually between UV and visible light region, and we have added the wavelength of irradiation in the revised manuscript.

Q6. Line 44/45 - Too many adjectives.

A. Thanks for your suggestions. We have revised the manuscript and polished the English language and grammar.

Q7. Line 46 - "Magnets" should be "magnetism".

A. Thanks. We have modified "Magnets" as "magnetism" in the revised manuscript.

Q8. Line 49 - "substantially" is unnecessary in this sentence.

A. Thanks. We have deleted this word in the revised manuscript.

Q9. Line 58 - $T_{1/2}$ arrows not defined.

A. Thanks for your advice. We have added the meaning of $T_{1/2}$ arrows as follows: " $T_{1/2\downarrow}$ and $T_{1/2\uparrow}$ represent the transition temperatures in the cooling and heating processes, respectively, during dc magnetic susceptibility measurements."

Q10. Line 61 - Reference missing.

A. Thanks for your advice. We have added the references in the revised manuscript.

Experimental section;

Q11. Yields only as percentages, no mass yields.

A. We thank you for the reviewer's valuable suggestions. We have added the mass yields in the experimental sections.

Q12. Section describing equipment used should go before description of synthesis.

A. We thank you for the reviewer's valuable suggestions. We have adjusted the section describing equipment before description of synthesis.

Q13. Line 100 - *via* is in italics but this is not consistently applied throughout manuscript.

A. We thank you for the reviewer's valuable suggestions. We have modified the word "*via*" in the format of italics and all of them have been changed in the revised manuscript.

Q14. Line 105/107 - inconsistent notation of X-ray wavelength used.

A. Thanks. the notation of X-ray wavelength is consistent in the revised manuscript now.

Q15. Line 112 - "was" should be "is".

A. Thanks. We have modified the word "was" as "is" in the revised manuscript. We also checked the grammar in the revised manuscript.

Q16. Line 149 - "totally" should be replaced with "could in total".

A. Thanks. We have modified this word in the revised manuscript.

Q17. Line 151 - "should" needs explaining and/or a reference.

A. Thanks. We have added two references in the revised manuscript.

50. Harada, J., Uekusa, H. & Ohashi, Y. X-ray Analysis of Structural Changes in Photochromic

Salicylideneaniline Crystals. Solid-State Reaction Induced by Two-Photon Excitation. *J. Am. Chem. Soc.* **121**, 5809–5810 (1999).

51. Li, P. X., Wang, M. S. & Guo, G. C. Two New Coordination Compounds with a Photoactive Pyridinium-Based Inner Salt: Influence of Coordination on Photochromism. *Cryst. Growth Des.* **16**, 3709–3715 (2019).

Q18. Line 154 - "ensure the amount..." sentence needs ending, is it to ensure the amount is increased/consumed etc?

A. Thanks for the kind advice from the reviewer. For the electron transfer photochromic materials, it is usually necessary to increase the time of light irradiation for ensuring the efficiency of electron transfer and the sufficiently photogenerated radicals, since photochromic reactions mainly happen in the surfaces of crystals. Base on this matter, we used ground powder samples and sufficient light time to ensure the sufficiently photogenerated radicals. To avoid the misunderstanding of the reviewers and readers, we revised the sentence as follows: "Thus, crystal samples of both compounds were ground to powder, and the light-irradiation time for photochromic experiments was increased to at least 180 min to ensure the sufficiently photogenerated radicals."

Q19. Line 191 - "may result in.." needs either evidence or a reference to back up this claim.

A. Thanks. We have added a reference related to a Mn-radical photochromic system (*Chem. Commun.* 2019, 55, 5631) in the revised manuscript.

Q20. Line 194 - Value of isolated high spin Mn^{2+} would be beneficial.

A. Thanks. We have added the χT value ($4.375 \text{ cm}^3 \text{ K mol}^{-1}$) of isolated high spin Mn^{2+} in the revised manuscript.

Q21. Line 211 - "orbital angular momentum contribution".

A. Thanks. We have changed this phrase in the revised manuscript.

Q22. Line 240 - "write formula" presumably is supposed to be a reference complex?

A. Thanks a lot. As the reviewer suggested, "write formula" was a reference complex, and we deleted this phrase to avoid the misunderstanding of readers.

Q23. Line 249, 250, 251 - there are no ESD values compared with later in the manuscript.

A. Thanks. We have added the ESD values in the revised manuscript.

Q24. Conclusion - DFT for complex **2** referred to again but no DFT for **2** in manuscript.

A. Thanks for the valuable suggestions. We have performed the DFT calculations for all the compounds including compound **2** and added the results in the conclusion part and supporting information.

Q25. Figure 1 - I think the use of capped sticks is limited here, ellipsoids would be better &/or inclusion of a chemdraw would help. Caption notes green is M^{2+} and also used to shade the M octahedra, doesn't help with clarity of image.

A. Thanks for the nice advices from the reviewer. As suggested, we have modified Figure 1 as follows and changed this picture in the revised manuscript.

Previous version

Revised version

Figure 1 The structure of compound **1**: (a) the molecular structure for compound **1** with thermal ellipsoids set at 50% probability; (b) the chain structure for **1**. The anthracene rings were shaded aquamarine with grey; (c) the H-bonding interactions for **1** between the chains. The green, grey-40%, red, blue and grey-25% colors represented Mn^{2+} , C, O, N and H atoms, respectively.

SI;

Q25. Figure S1 is described as "The molecular structure for all the compounds" whereas the main text refers to figure S1 as the structure of complex **1**.

A. Thank you for your valuable suggestions. We have modified the figures and added the structures of **2–4** in SI.

Previous version

Revised version

Figure S2. The 1D structure for compounds **2** (a), **3** (b) and **4** (c) with thermal ellipsoids set at 50% probability. The green, grey-40%, red and blue colors represented M^{2+} ($M = \text{Zn}, \text{Ni}$ and Co for **2**, **3** and **4**, respectively), C, O and N atoms, respectively. H atoms are omitted for clarity.

Q26. Figure S9 states "The color changes of compound **3** and **4**" but it is very difficult to discern any colour changes in either a) or b).

A. Thank you for your valuable suggestions. We are sorry for misleading the reviewer. Compounds **3** and **4** didn't occur any photochromism, and no color changes were observed during Xe lamp light irradiation. Our original intention was to compare the variations of magnetic couplings under the photogenerated radicals or not after light irradiation for these compounds and we had stated this phenomenon in the initial manuscript. However, our description induced the misunderstanding of the reviewer. To not mislead the reviewers and the readers, we modified the caption of Figure S9 as follows: "The color of compound **3** (a) and **4** (b) before and after Xe lamp light irradiation. As seen in the pictures, no observable photochromic phenomenon occurred in these two compounds.". Furthermore, the time-dependent UV-Vis and photoluminescence spectra upon light irradiation remained unchanged, this also demonstrated that there was no photochromism occurred in compounds **3** and **4**. We also modified the sentences about the description of **3** and **4** in the revised manuscript and supporting information.

Q27. Figure S10 has IR for both complex **3** & **4**, but not explicitly clear which is which,

compared to Figure S11 where the authors have noted in the caption which UV-Vis spectrum belongs to which.

A. Thank you for your valuable suggestions. We have modified the caption of Figure S10 and the revised IR curves are as follows.

Figure S14. IR plots for compounds 3 (a) and 4 (b) at 293 K.

Q28. A greater degree of consistency in the manuscript and SI would be appreciated.

A. Thank you for your valuable advices. As the reviewer suggested, we have checked the manuscript and now the revised manuscript and SI are consistent with each other.

Whilst the work has clearly been conducted to a rigorous standard including a good deal of data collection, I would not recommend that this manuscript be published in Nat. Commun. for the reasons described. Instead, I would suggest the authors submit to a different journal.

Thanks very much again for the reviewer to review and improve our work. According to the comments from the reviewer, we have drastically revised and improved our manuscript, all the concerns have been fully addressed throughout the manuscript. Some other minor revisions have also been made in the revised manuscript and supporting information. We hope the revised manuscript could meet the expectations of the reviewer for publication in *Nature Communications*.

Reviewer #2 (Remarks to the Author):

I think this is a nice paper of valid scientific results, however I have small issues regarding the crystallographic part:

Response: We are grateful for the kind and valuable comments from reviewer and the suggestions have enabled us to improve our work. Here is our point-by-point response to the comments raised by the reviewer. Thanks very much for reviewing and improving our work.

Q1. Every single structure shows the same ADDSYM alert, sometimes at B level and sometimes at C level, regarding the possible wrong space group choice. I would like to see the authors dealing with this at least with a comment.

A. We are grateful for the kind and valuable comments from the reviewer. As the reviewer suggested, the structures showed the same ADDSYM alert. However, according to the results of ADDSYM in the PLATON software, there is no higher symmetry in the structure of all the compounds. Furthermore, we tried but failed to solve the structures via $C2/m$ or $I2/m$ space group. The follow pictures are the results from the XPREP process (compound **1** as the example):

Figure R1 the screenshots of the XPREP process for **1** when selected the P crystal lattice.

```

Current dataset: 20201221167.hkl      Wavelength: 0.71073 Chiral: ?
-----
Original cell:  7.447 14.085 11.361  90.00 107.56  90.00  Vol: 1138.1
             h k l      0.00  0.01  0.00  0.00  0.01  0.00  Lattice: C
-----
Current cell:  7.447 14.085 11.361  90.00 107.56  90.00  Vol: 1138.1
-----
Matrix: 1.0000 0.0000 0.0000 0.0000 1.0000 0.0000 0.0000 0.0000 1.0000
-----
Crystal system: Monoclinic      Lattice: C

[S] Determine SPACE GROUP
[C] Must be CHIRAL (sample is optically active)
[M] NOT NECESSARILY chiral (eg. may be racemate)
[I] INPUT known space group
[E] EXIT to main menu or [Q] QUIT program

Select option [S]:

[A] Triclinic, [M] Monoclinic, [O] Orthorhombic, [T] Tetragonal,
[R] Trigonal/Rhombohedral, [C] Cubic or [E] EXIT

Select option [M]:

Lattice exceptions: P  A  B  C  I  F  Obv  Rev  All
N (total) =          0  4454  4459  4471  4483  4493  5550  5946  8539
N (int>sigma) =       0  2057  3081  3084  3020  4112  4054  4048  4094
Mean intensity =     0.0  1.7 12.1 12.0 11.6  8.6 11.7 11.9 11.7
Mean int/sigma =     0.0  4.1 16.9 16.6 16.1 13.2 16.2 16.4 16.2

Lattice type [P, A, B, C, I, F, Obv., R (rev. show. on hex. axes)]

Select option [F]: C

Mean [I^2-1] = 1.052 [expected .968 centrosym and .736 non-centrosym]

Systematic absence exceptions:
-----
-c-
h
h k l
 0.1
<I/*> 1.0

Identical indices and Friedel opposites combined before calculating R(sym)

Option Space Group No. Type Axes CSD R(sym) N(eq) Syst. Abs. CFOM
[A] C2/c # 19 centro 1 3496 0.017 1579 1.0 / 16.2 1.99
[B] Cc # 9 non-cent 1 546 0.017 1579 1.0 / 16.2 12.83

Select option [A]:

```

Figure R2 the screenshots of the XPREP process for **1** when selected the *C* crystal lattice.

When we selected the *P* crystal lattice, the structure was easily solved. However, when we selected the *C* crystal lattice, the structure was failed to solve. For all the compounds, including **1** before and after light irradiation, they are all pseudo-symmetric, however, this ADDSYM alert does not contribute to the magnetic transitions. Actually, magnetic bistability with the extremely wide thermal hysteresis is originated from variations of the Mn²⁺ ions and photogenerated radicals, induced by the rotation of the anthracene rings changed the Mn1–O2–C8 angle and coordination geometries of the Mn²⁺ center.

Q2. Comparing powder pattern of different compounds simply “by eye” with a calculated one from Mercury is not the best way of doing. Instead, I strongly recommend a Rietveld refinement between the experimental pattern and the one from the single crystal data.

A. Thanks for the kind suggestions from the reviewer. As the reviewer suggested, we added a Rietveld refinement between the experimental pattern and the one from the single crystal data for all the compounds. The figures have been added in the revised SI and showed as follows:

Figure S4. Le Bail method fit to PXRD data for compounds **1** (a) and **1a** (b).

Figure S5. Le Bail method fit to PXRD data for compounds **2** (a) and **2a** (b).

Figure S6. Le Bail method fit to PXRD data for compound **3**.

Figure S7. Le Bail method fit to PXRD data for compound **4**.

Once these issues have been dealt with, I recommend this manuscript for publication.

A. Thanks again for improving our work, and we hope the revised manuscript could address the reviewers' comments and meet the expectations of the reviewer for publication in *Nature Communications*.

Reviewer #3 (Remarks to the Author):

This work for the first time achieved the magnetic thermal hysteresis loop with the temperature width of 177 K in the radical-actuated photochromic materials. It is very interesting for the readers of *Nature Commun.* I recommend it to be published in *Nature Commun.* after considering the following questions.

Response: We are grateful for the kind and valuable comments from reviewer and the suggestions have enabled us to improve our work. Here is our point-by-point response to the comments raised by the reviewer. Thanks very much for reviewing and improving our work.

Q1. They indicate there are strong magnetic couplings between the photogenerated radicals and Mn^{2+} . However, I don't find the exchange coupling constant J between them. I suggest authors to calculate J using DFT. The computational details can be found in *Inorg. Chem.* 2021,60,1007.

A. Thanks for the valuable suggestions from the reviewer. As the reviewer suggested, we calculated the exchange coupling constant J using DFT calculations. According to the reference *Inorg. Chem.* 2021, 60, 1007, Zhang *et al.*⁵⁷ had succeeded in using the popular hybrid functional O3LYP⁵⁸ to obtain the closest exchange coupling constants between Co^{II} and radicals to the experiment. To obtain the exchange coupling constants J between Mn^{2+} and radicals, we firstly extracted a fragment of ADC- Mn^{2+} -ADC (Figure 4a) from 1D chain of **1a**, and then used O3LYP with ORCA 5.0.2⁵⁹ to calculate them. Def2-TZVP with auxiliary coulomb fitting basis set of SARC/J⁶⁰ was used for all atoms, and zero order regular approximation (ZORA) was used for the scalar relativistic effect in the calculation. The tight grid of DEFGRID3 and tight convergence criteria were selected to ensure that the results are well converged with respect to technical parameters. For simplification, only the main nearest neighboring Mn^{2+} -ADC exchange interactions were considered.

We firstly calculated the energy of the high-spin (HS) state ($S_{\text{HS}} = 1/2+1/2+5/2 = 7/2$), and then flipped the spins on all of atoms except for Mn^{2+} to obtain the energy of the low-spin (LS) state ($S_{\text{LS}} = -1/2-1/2+5/2 = 3/2$). From the spin density distributions in Figure 4b and 4c, we find that the spin densities in the HS and LS states are mainly distributed over Mn^{2+} and only one ADC due to omitting the nearest $\text{Mn}^{2+}(\text{H}_2\text{O})_2(\text{DMF})_2$ around the other ADC.

Figure 4 (a) Extracted fragment of ADC-Mn²⁺-ADC from compound **1a**. H atoms are omitted for clarity; Spin density distributions for the extracted fragment in the HS (b) and LS (c) states (blue and red regions indicate positive and negative spin populations, respectively; the isodensity surface represented corresponds to a value of 0.002 e⁻ bohr⁻³).

The corresponding spin values on Mn²⁺ and one ADC in the HS state are 4.896 and 2.019, respectively, and 4.889 and -1.970 in the LS state. Thus, we only need to consider the exchange coupling between Mn²⁺ and one ADC for our created fragment. The spins of Mn²⁺ and one ADC are 5/2 and 1, respectively. According to Hamiltonian: $\hat{H} = -2J_{Mn-ADC} \hat{S}_{Mn} \hat{S}_{ADC}$. The Mn²⁺-ADC coupling constant J_{Mn-ADC} was obtained through eqs 1 and 2 using the spin-projected and non-spin-projected approach, respectively.⁶¹⁻⁶³

$$J_{Mn-ADC} = \frac{E_{LS} - E_{HS}}{10} \quad (\text{spin projection}) \quad (1)$$

$$J_{Mn-ADC} = \frac{E_{LS} - E_{HS}}{12} \quad (\text{non-spin projection}) \quad (2)$$

The calculated J_{Mn-ADC} according to eqs 1 and 2 are -11.4 and -9.5 cm⁻¹, respectively. The negative J_{Mn-ADC} values suggest the exchange couplings between Mn²⁺ and ADC are antiferromagnetic for 1D chain of **1a**.

57. Lu, F., Li, J. X., Guo, W. X., Wang, B. L. & Zhang, Y. Q. Origin of Magnetic Relaxation Barriers in a Family of Cobalt(II)-Radical Single-Chain Magnets: Density Functional Theory and *Ab Initio* Calculations. *Inorg. Chem.* **60**, 1007–1015 (2021).
58. Handy, N. C. & Cohen, A. J. Left-Right Correlation Energy. *Mol. Phys.* **99**, 403–412 (2001).
59. Neese, F. ORCA—an *ab initio*, Density Functional and Semiempirical Program Package, version 5.0.2; Max-Planck institute for bioinorganic chemistry: Mülheim an der Ruhr, Germany. (2021).
60. Rolfes, J. D., Neese, F. & Pantazis, D. A. All-Electron Scalar Relativistic Basis Sets for the Elements Rb–Xe. *J. Comput. Chem.* **41**, 1842–1849 (2020).

61. Ruiz, E., Rodríguez-Forteza, A., Cano, J., Alvarez, S. & Alemany, P. About the Calculation of Exchange Coupling Constants in Polynuclear Transition Metal Complexes. *J. Comput. Chem.* **24**, 982–989 (2003).
62. Noodleman, L. & Case, D. A. Density-Functional Theory of Spin Polarization and Spin Coupling in Iron-Sulfur Clusters. *Adv. Inorg. Chem.* **38**, 423–470 (1992).
63. Bencini, A. & Totti, F. A Few Comments on the Application of Density Functional Theory to the Calculation of the Magnetic Structure of Oligo-Nuclear Transition Metal Clusters. *J. Chem. Theory Comput.* **5**, 144–154 (2009).

Q2. They conclude the thermal hysteresis of **1a** was due to the variations of magnetic couplings between Mn^{2+} and photogenerated diradicals and interchain H-bonding interactions. **3a** and **4a** also have the photogenerated radicals, but why they have no thermal hysteresis. Please give more detail analyses.

A. Thanks for the kind advices from the reviewer. In our initial manuscript, we also performed the photochromic behavior of compounds **3** and **4**, while no obvious photochromism was observed after Xe lamp light irradiation, which was demonstrated by the UV-Vis and luminescence spectra in our initial manuscript. To facilitate the comparison of the variations of magnetic couplings for all the compounds, we also named the compounds **3** and **4** as **3a** and **4a** after light irradiation for clarity (in fact, the molecular and electronic structures are the same because of no photochromism occurred in compounds **3** and **4**). However, this caused ambiguity and we are sorry for misleading the reviewers. As a result, without photogenerated radicals, compounds **3** and **4** didn't show magnetic thermal hysteresis loops. Furthermore, we also deleted the syntheses of compounds **3a** and **4a**, and modified some sentences to not mislead the reviewers and readers in the revised manuscript.

Finally, thanks very much again for reviewing and improving our work, we hope the revised manuscript could meet the expectations of the reviewer for publication in *Nature Communications*.

Reviewer #4 (Remarks to the Author):

The paper reports a very important result on the obtaining of an electron transfer compound with an exceptionally large thermal hysteresis loop. Several compounds have been prepared and only one exhibits this remarkable behavior. Each compound has been deeply characterized to evidence and explain the origin of such large bistability. Even if the characterizations are mainly convincing, some information are lacking. I would have 3 major points to express before allowing this paper to be published in Nature Comm.

Response: We are grateful for the kind and valuable comments from reviewer and the suggestions have enabled us to improve our work. Here is our point-by-point response to the comments raised by the reviewer. Thanks very much for reviewing and improving our work.

Q1. If the electron transfer is between a coordinated DMF and the anthracene units why compound **2** that shows the ET does not exhibit any hysteresis? The explanation concerning the absence of this ET in **3** and **4** focused on the absorption bands (page 78, line 143) seems also incomplete. What is the ET pathway? Through the metal center? Through space? What the impact of the metal on the redox potential that can favor or prevent the ET to occur?

A. Thanks for the valuable suggestions from the reviewer. In our work, magnetic bistability with large thermal hysteresis for **1a** is due to the variations of magnetic couplings between Mn^{2+} ions and photogenerated radicals originated from the changes in the Mn1–O2–C8 angle. Therefore, compound **2** without paramagnetic metal ions and compounds **3** and **4** without photogenerated radicals can't exhibited such thermal hysteresis.

For all the isostructural compounds, the obvious difference is the metal center. As a result, the occurrence of electron transfer photochromism is relevant to the metal ions. For **1** and **2**, the $3d$ orbitals in these metals are half filled ($3d^5$) or fully filled ($3d^5$), they are in the relatively stable states. However, the $3d$ orbitals in these metals are $3d^8$ and $3d^7$, respectively, they easily occur the ligand-to-metal electron transfer process when theses metal ions coordinate with organic ligands, which may in turn hinder the occurrence of ligand-to-ligand electron transfer photochromism. Furthermore, $d-d$ transitions existed in compounds **3** and **4** (Figure S15), this absorption is overlapped with the photogenerated radicals, and this overlap should also prevent

the appearance of the electron transfer photochromism.

To explore the impact of the redox potential of metal on the photochromic behavior, we measured the Cyclic voltammograms of **1–4** during the reduction process, since the radicals may oxidize the metal ions during the metal-assisted LLCT process. As seen in Figure R3, this redox process for all the compounds is occurred in about 0.2 V, this suggested that electron transfer photochromic behavior is irrelevant to the metal on the redox potential.

Figure R3 LSV curves for **1–4** in Ar saturated 0.1 M Na₂SO₄ electrolyte with a scan rate of 50 mV/s.

For reliably explain the electron transfer process, we extracted a fragment of ADC²⁻-Mn²⁺-ADC²⁻ from 1D chain and use the Gaussian 09 program and the basis set B3LYP/6-311G(d) method to calculate these series of compounds. The DFT calculations indicate that the electron transfer process is mainly occurred in the ADC²⁻ ligand, and the localization of HOMO and LUMO suggests the HOMO-LUMO transition with an intraligand electron transfer from carboxyl groups to anthracene motifs. However, the DFT results don't mean that the electron transfer occurs in only one ADC ligand, the adjacent ADC ligands coordinated with the same metal center should also participate in this electron transfer process, due to the fact that the sole ADC ligand could appear self-photochromic behavior and the constructed isostructural compounds showed different photochromic behavior. Actually, the metal ions should participate in the electron transfer process, and this metal-assisted ligand-to-ligand electron transfer has been widely researched in many works (*Nature Commun.* 2020, 11, 1179; *J. Am. Chem. Soc.* 2021, 143, 2232; *Coord. Chem. Rev.* 2022, 452, 214304).

Q2. I would strongly suggest to perform and show the DFT calculations for all the compounds to compare the ET pathway and efficiency. If the metal is not involved, it should not depend on its nature. Moreover, what are the LUMO and HOMO features in the photo-excited states? Since the XRD data are available, it might be interesting to perform the same calculations. This would help to understand the magnetic properties.

A. Thanks very much for the valuable suggestions from reviewer. The most important highlight in our work is the first realization of magnetic bistability with a large thermal hysteresis of 177 K in the electron transfer photochromic materials. However, as the reviewer suggested, the ET pathway and efficiency in the electron transfer process is vital and a thorough DFT/computational analysis for all the compounds is indispensable for improving our work. We also modified the description of the abstract, text and conclusion. To reliably explain the electron transfer process, we extracted a fragment of $\text{ADC}^{2-}\text{-Mn}^{2+}\text{-ADC}^{2-}$ from the 1D chain and used the Gaussian 09 program and the basis set B3LYP/6-311G(d) method to calculate these series of compounds. For **1**, the DFT calculations indicated that the electron transfer process is mainly occurred in the ADC^{2-} ligand (Figure S25), and the localization of HOMO and LUMO suggested the HOMO-LUMO transition with an intraligand electron transfer from carboxyl groups to anthracene motifs. For the light irradiated sample **1a**, the DFT results also concluded that electron distributions are mainly located on the ADC ligands (Figure S26), and the photogenerated radicals should be delocalized in the ADC units. After light irradiation, the Mn^{2+} ions coordinated with the photogenerated ADC^{\bullet} radical analogs, further inducing the photomagnetic behavior with magnetic bistability in a large temperature region. We also modified the manuscript as follows and added the DFT calculations in the supporting information. “The transfer pathway for **1** was confirmed by density functional theory (DFT) calculations, and the spatial distributions of the highest occupied molecular orbital (HOMO) and lowest unoccupied molecular orbital (LUMO) were calculated using the Gaussian 09 program and the basis set B3LYP/6-311G(d) method. As shown in Figures 2g, 2h, and S25, electrons were mainly distributed in the ADC^{2-} ligands, indicating that electron-transfer photochromism in **1** is mainly originated from the photoactive H_2ADC ligands. However, the electron distribution in the HOMO was dominantly located on the carboxyl groups, whereas that in LUMO was

dominant on the anthracene rings. The localization of HOMO and LUMO suggests the HOMO–LUMO transition is attributed to intraligand electron transfer from carboxyl groups to anthracene motifs.⁵⁶ For light irradiated **1a**, DFT calculations also revealed that electron distribution is mainly located on the ADC ligands (Figure S26), and the photogenerated radicals are delocalized in the ADC units. Thus, the photogenerated ADC[•] radical species coordinate with Mn²⁺ centers to trigger strong magnetic couplings and induce photomagnetic behavior.⁵⁴ Furthermore, the difference in the frontier molecular orbitals from HOMO–4 to LUMO+4 for **2–4** was calculated, and electrons were distributed in the ADC²⁻ ligands in all the compounds (Figures S27–S29). However, the DFT result shows that electron transfer occurs not only in one ADC ligand, but also in the adjacent ADC ligands coordinate with the same metal center, because the sole ADC ligand exhibits self-photochromic behavior, and the constructed isostructural compounds show different photochromic behavior. The metal ions should also participate in the electron transfer process. This metal-assisted ligand-to-ligand electron transfer has been widely studied in many works.^{39,47*}

Figure S25. HOMO, HOMO–1, HOMO–2, HOMO–3, HOMO–4, LUMO, LUMO+1, LUMO+2, LUMO+3 and LUMO+4 frontier molecular orbitals of compound **1** obtained by DFT calculations.

Figure S26. HOMO, HOMO–1, HOMO–2, HOMO–3, HOMO–4, LUMO, LUMO+1, LUMO+2, LUMO+3 and LUMO+4 frontier molecular orbitals of compound **1a** obtained by DFT calculations.

Figure S27. HOMO, HOMO-1, HOMO-2, HOMO-3, HOMO-4, LUMO, LUMO+1, LUMO+2, LUMO+3 and LUMO+4 frontier molecular orbitals of compound **2** obtained by DFT calculations.

Figure S28. HOMO, HOMO-1, HOMO-2, HOMO-3, HOMO-4, LUMO, LUMO+1, LUMO+2, LUMO+3 and LUMO+4 frontier molecular orbitals of compound **3** obtained by DFT calculations.

Figure S29. HOMO, HOMO-1, HOMO-2, HOMO-3, HOMO-4, LUMO, LUMO+1, LUMO+2, LUMO+3 and LUMO+4 frontier molecular orbitals of compound **4** obtained by DFT calculations.

Furthermore, the kinetics of the coloration for **2** was also analyzed in order to estimate the rate of photochromism, using function $R^{\lambda_{\max}}(t) = a/(bt + 1) + R^{\lambda_{\max}}(\infty)$ to fit peak of diffuse reflection versus time curve. The relative parameters are listed in the Figure S11b, wherein, the constant a and b are fitting results, $R^{\lambda_{\max}}(\infty)$ is the peak value of diffuse reflection spectrum after complete irradiation. Furthermore, substituting $R(t) = [R(0) + R(\infty)]/2$ into the function gives the value of half-time $t_{1/2}$ about 12.31 min. Compared with $t_{1/2}$ of 9.20 min for **1**, the photochromic rate for **2** indicates a relatively slower coloration process, which could be observed in the time-dependent photoluminescence spectra. The efficiency of both photochromic compounds has been compared in the revised manuscript.

Figure S11. (a) Time-dependent UV-vis diffuse-reflectance spectra of **2** upon irradiation, which were obtained by transforming the absorption spectra; b) The plot of relative intensity of time-dependent UV-Vis spectra at 450 nm upon light irradiation. The red solid line presented the fitted curve. The kinetics of the coloration for **2** was analyzed in order to estimate the rate of photochromism, using function $R^{\lambda_{\max}}(t) = a/(bt + 1) + R^{\lambda_{\max}}(\infty)$ to fit peak of diffuse reflection versus time curve. The relative parameters are listed in the Figure S11b, wherein, the constant a and b are fitting results, $R^{\lambda_{\max}}(\infty)$ is the peak value of diffuse reflection spectrum after complete irradiation. Furthermore, substituting $R(t) = [R(0) + R(\infty)]/2$ into the function gives the value of half-time $t_{1/2}$ about 12.31 min.

Q3. Concerning the magnetic properties and cooperativity, it would be interesting to use the DFT calculation to access the coupling constant between the radical and the metal, and compare it with data fitting. Moreover, it is not clear why in **1a** the H bonding network leads to such large hysteresis. Is cooperativity really bigger, and the only explanation? Again why **2a** does not show any thermally-induced ET?

A. Thanks for the valuable suggestions from the reviewer. As the reviewer suggested, we calculated the exchange coupling constant J using DFT calculations. According to the reference *Inorg. Chem.* 2021, 60, 1007, Zhang *et al.*⁵⁷ used the popular hybrid functional O3LYP⁵⁸ to obtain the closest exchange coupling constants between Co^{II} and radicals. To obtain the exchange coupling constant J between Mn^{2+} and radicals, we first extracted a fragment of $\text{ADC}^{2-}-\text{Mn}^{2+}-\text{ADC}^{2-}$ (Figure 4a) from the 1D chain of **1a** and then used O3LYP with ORCA 5.0.2⁵⁹ to calculate them. Def2-TZVP with auxiliary coulomb fitting basis set of SARC/J⁶⁰ was used for all atoms, and the zero-order regular approximation was employed for the scalar relativistic effect in the calculation. The tight grid of DEFGRID3 and tight convergence criteria were selected to ensure that the results converged well with respect to the technical parameters.

For simplification, only the nearest neighboring Mn^{2+} - ADC^{2-} exchange interactions are considered.

Figure 4 (a) Extracted fragment of $\text{ADC-Mn}^{2+}\text{-ADC}$ from compound **1a**. H atoms are omitted for clarity; Spin density distributions for the extracted fragment in the HS (b) and LS (c) states (blue and red regions indicate positive and negative spin populations, respectively; the isodensity surface represented corresponds to a value of $0.002 \text{ e}^- \text{ bohr}^{-3}$)

First, we calculated the energy of the HS state ($S_{\text{HS}} = 1/2 + 1/2 + 5/2 = 7/2$) and then flipped the spins on all the atoms except for Mn^{2+} to obtain the energy of the low-spin (LS) state ($S_{\text{LS}} = -1/2 -1/2 + 5/2 = 3/2$). The spin-density distributions (Figures 4b and 4c) showed that the spin densities in the HS and LS states are mainly distributed over Mn^{2+} and only one ADC^{2-} because the nearest $\text{Mn}^{2+}(\text{H}_2\text{O})_2(\text{DMF})_2$ around the other ADC^{2-} was omitted. The corresponding spins on Mn^{2+} and one ADC^{2-} are 4.896 and 2.019 in the HS state and 4.889 and -1.970 in the LS state, respectively. Thus, only the exchange coupling between Mn^{2+} and one ADC^{2-} should be considered for the created fragment. The spins of Mn^{2+} and one ADC^{2-} are $5/2$ and 1 , respectively. According to Hamiltonian, $\mathbf{H} = -2J_{\text{Mn-ADC}} \mathbf{S}_{\text{Mn}} \mathbf{S}_{\text{ADC}}$. The $\text{Mn}^{2+}\text{-ADC}^{2-}$ coupling constant $J_{\text{Mn-ADC}}$ was obtained based on Eqs. 1 and 2 using the spin- and non-spin-projected approaches, respectively.⁶¹⁻⁶³

$$J_{\text{Mn-ADC}} = \frac{E_{\text{LS}} - E_{\text{HS}}}{10} \quad (\text{spin projection}) \quad (1)$$

$$J_{\text{Mn-ADC}} = \frac{E_{\text{LS}} - E_{\text{HS}}}{12} \quad (\text{non-spin projection}) \quad (2)$$

$J_{\text{Mn-ADC}}$ calculated using Eqs. 1 and 2 are -11.4 and -9.5 cm^{-1} , respectively. The negative $J_{\text{Mn-ADC}}$ values suggest exchange couplings between Mn^{2+} and ADC^{2-} are antiferromagnetic for the 1D chain of **1a** constant with the Curie-Weiss fitting results.

We also analyzed the cooperativity (π - π stacking and interchain H-bonding interactions) for both compounds. There are only interchain H-bonding interactions are observed in the crystal structures. The interchain distances, interchain H-bonding interactions and volume in the crystal shrank after photogeneration of radicals, and the shrinkages are observed in many other electron transfer photochromic compounds (*Angew. Chem. Int. Ed.* 2017, 56, 14458; *J. Am. Chem. Soc.* 2020, 142, 2682). Furthermore, the H-bonding interactions has been demonstrated to play an important role in the magnetic transition and thermal hysteresis in the spin transition materials (*Chem. Soc. Rev.* 2011, 40, 4119; *Chem. Lett.* 2014, 43, 1178; *Chem. Eur. J.* 2021, 27, 740). In this work, the H-bonding interactions in **1a** connected the photogenerated radical- Mn^{2+} single chains to an extended 2D structure, and this enhanced cooperativity may be also a reason for the hysteresis (*Chem. Eur. J.* **18**, 2012, 15230). However, the large thermal hysteresis in our work is still originated from the variations of magnetic couplings between photogenerated radicals and Mn^{2+} ions, which was supported by variable temperature crystal data, magnetic curves and DFT calculations.

In general, the radical species may also exhibit magnetic bistability with thermal hysteresis, but this phenomenon usually occur in the system of reversible dimerization of organic π radicals. However, we didn't observe the dimerization of π radicals in our work, consequently, **2** showed no thermal hysteresis even the generation of radicals after light irradiation. For all the compounds, radicals could be generated in **1** after light irradiation and couple with Mn^{2+} ions; for **2**, the spin carrier is just photogenerated radicals; while for **3** and **4**, there are no observable radicals even the existence of anisotropic Ni^{2+} ion and orbital angular momentum contribution of Co^{2+} ion. Evidently, the most obvious difference for all these compounds is the presence or absence of magnetic coupling between photogenerated radicals and metal ions. As a result, the observed thermal hysteresis loop in **1** is mainly due to the magnetic couplings between photogenerated radicals and Mn^{2+} ions.

57. Lu, F., Li, J. X., Guo, W. X., Wang, B. L. & Zhang, Y. Q. Origin of Magnetic Relaxation Barriers in a Family of Cobalt(II)–Radical Single-Chain Magnets: Density Functional Theory and *Ab Initio* Calculations. *Inorg. Chem.* **60**, 1007–1015 (2021).
58. Handy, N. C. & Cohen, A. J. Left-Right Correlation Energy. *Mol. Phys.* **99**, 403–412 (2001).
59. Neese, F. ORCA—an *ab initio*, Density Functional and Semiempirical Program Package, version 5.0.2; Max-Planck institute for bioinorganic chemistry: Mülheim an der Ruhr, Germany. (2021).
60. Rolfes, J. D., Neese, F. & Pantazis, D. A. All-Electron Scalar Relativistic Basis Sets for the Elements Rb–Xe. *J. Comput. Chem.* **41**, 1842–1849 (2020).
61. Ruiz, E., Rodríguez-Fortea, A., Cano, J., Alvarez, S. & Alemany, P. About the Calculation of Exchange Coupling Constants in Polynuclear Transition Metal Complexes. *J. Comput. Chem.* **24**, 982–989 (2003).
62. Noodleman, L. & Case, D. A. Density-Functional Theory of Spin Polarization and Spin Coupling in Iron-Sulfur Clusters. *Adv. Inorg. Chem.* **38**, 423–470 (1992).
63. Bencini, A. & Totti, F. A Few Comments on the Application of Density Functional Theory to the Calculation of the Magnetic Structure of Oligo-Nuclear Transition Metal Clusters. *J. Chem. Theory Comput.* **5**, 144–154 (2009).

Q4. Finally, I think the English might be improved and refs 6 and 7 changed to more general ones related to SCO like the Gutlich and Halcrow books.

A. Thanks for your kind advices. As the reviewer suggested, we ask a colleague to review our manuscript (Hiroki Oshio in Dalian University of Technology) and further use an English language editing service (Enago) to polish and improve our manuscript, and now the revised manuscript would strongly benefit from English language editing. We also modified refs 6 and 7 as Gutlich and Halcrow books followed the reviewer’s advice.

6. Gütlich, P. & Goodwin, H. A. Spin Crossover in Transition Metal Compounds I–III. *Top. Curr. Chem.* **233-235** (2004).

7. Halcrow, M. A. Spin-Crossover Materials: Properties and Applications. *John Wiley & Sons* (2013).

Finally, thanks very much again for reviewing and improving our work, and we hope the revised manuscript could meet the expectations of the reviewer for publication in *Nature Communications*.

REVIEWER COMMENTS

Reviewer #1 (Remarks to the Author):

I would like to commend the authors for the amount of work they have done, both with the original submission but particularly in addressing the comments I originally raised. The resubmitted manuscript is now, I believe, is very good and highly deserving of being published in Nature Communications.

Reviewer #2 (Remarks to the Author):

I am happy with the updates and the explanations that the authors have provided regarding the crystallographic part. I can now recommend the manuscript for publication

Reviewer #3 (Remarks to the Author):

It's OK.

Reviewer #4 (Remarks to the Author):

The paper by Hu et al has been deeply improved, accounting for the various comments from the reviewers. It may be published after un final reading.
I would change in the title "electron-transfer photochromism" by "photo-induced electron transfer" since photochromism is a consequence of the ET not the cause.

Response to Reviewer Comments

Reviewer: 1

I would like to commend the authors for the amount of work they have done, both with the original submission but particularly in addressing the comments I originally raised. The resubmitted manuscript is now, I believe, is very good and highly deserving of being published in Nature Communications.

Response: We are grateful for the kind and valuable comments from the reviewer. Thanks very much for reviewing and improving our work.

Reviewer #2 (Remarks to the Author):

I am happy with the updates and the explanations that the authors have provided regarding the crystallographic part. I can now recommend the manuscript for publication.

Response: We are grateful for the kind and valuable comments from the reviewer. Thanks very much for reviewing and improving our work.

Reviewer #3 (Remarks to the Author):

It's OK.

Response: We are grateful for the kind and valuable comments from the reviewer. Thanks very much for reviewing and improving our work.

Reviewer #4 (Remarks to the Author):

The paper by Hu et al has been deeply improved, accounting for the various comments from the reviewers. It may be published after un final reading.

I would change in the title "electron-transfer photochromism" by "photo-induced electron transfer" since photochromism is a consequence of the ET not the cause.

Response: We are grateful for the kind and valuable comments from the reviewer and the suggestions have enabled us to improve our work. As the reviewer suggested, we revised the title as “Achieving Large Thermal Hysteresis of an Anthracene-Based Manganese(II) Complex via Photo-Induced Electron Transfer”. Thanks very much for reviewing and improving our work.